# A 20-year satellite-reanalysis-based climatology of extreme precipitation characteristics over the Sinai Peninsula

Mohsen Soltani[1,2], Bert Hamelers[1,3], Abbas Mofidi[4], Christopher G. Fletcher[2], Arie Staal[5], Stefan C. Dekker[5], Patrick Laux[6,7], Joel Arnault[6], Harald Kunstmann[6,7], Ties van der Hoeven[8], Maarten Lanters[8]

[1] Natural Water Production Theme, European Centre of Excellence for Sustainable Water Technology (Wetsus), Leeuwarden, Netherlands
[2] Department of Geography and Environmental Management, University of Waterloo, Waterloo, Canada
[3] Sub-department of Environmental Technology, Wageningen University, Wageningen, Netherlands
[4] Department of Geography, Ferdowsi University of Mashhad, Mashhad, Iran
[5] Department of Environmental Sciences, Copernicus Institute, Utrecht University, Utrecht, Netherlands
[6] Institute of Meteorology and Climate Research, Karlsruhe Institute of Technology, Garmisch-Partenkirchen, Germany
[7] Institute of Geography, University of Augsburg, Augsburg, Germany
[8] The Weather Makers B.V., Burgemeester Loeffplein, 'S-Hertogenbosch, Netherlands
**Correspondence**: Mohsen Soltani (mohsen.soltani@uwaterloo.ca)

## Abstract

Extreme precipitation events and associated flash floods caused by the synoptic cyclonic-systems have profound impacts on society and environment particularly in arid regions. This study brings forward a satellite-reanalysis-based approach to quantify extreme precipitation characteristics over the Sinai Peninsula (SiP) in Egypt, from a statistical-synoptic perspective for the period of 2001-2020. With a multi-statistical-approach developed in this research, SiP's wet and dry periods are determined. Using satellite observations of precipitation and a set of derived precipitation indices, we characterize the spatiotemporal variations of extreme rainfall climatologies across SiP. Then, using the reanalysis datasets, synoptic systems responsible for the occurrence of extreme precipitation events along with the major tracks of cyclones during the wet and dry periods are described. Our results indicate that trends and spatial patterns of the rainfall events across the region are inconsistent in time and space. Highest precipitation percentiles (~20 mm/month), frequencies (~15 days/month with rainfall ≥10 mm/day), standard deviations (~9 mm/month), and monthly ratios (~18%) are estimated in north/northeastern parts of the region during the wet period especially in early winter; also, a substantial below-average precipitation condition (drier trend) is clearly observed in most parts except for the south. The Mediterranean cyclones accompanied by the Red Sea and Persian Troughs are responsible for the majority of extreme rainfall events year-round. A remarkable spatial relationship between SiP's rainfall and the atmospheric variables of sea level pressure, wind direction and vertical velocity is found. A cyclone-tracking analysis indicates that 125 cyclones (with rainfall ≥10mm/day) formed within, or transferred to, the Mediterranean basin and precipitated over SiP during wet periods, compared to 31 such cyclones during dry periods. It is estimated around 15% of cyclones with sufficient rainfall >40mm/day would be capable of leading to flash-floods during the wet period. This study, therefore, sheds new light on the extreme precipitation characteristics over SiP and its association with dominant synoptic-scale mechanisms over the eastern Mediterranean region.

## 1 Introduction

Extreme precipitation events can have fundamental impacts on society and human wellbeing by causing mortality (Trenberth *et al.,* 2007; Toreti *et al.,* 2010; Wannous and Velasquez 2017; Charlton-Perez *et al.,* 2019), and by causing property and ecological damages (Zhang *et al.,* 2005; IPCC, 2013; Nastos *et al.,* 2013; Boucek *et al.,* 2016). Precipitation extremes are realized as one of the severest natural disasters, among many others (Arnous and Omar 2018). Nevertheless, these events are vital for the water resources of the region especially in the water-limited environments (Peleg et al., 2012; Givati et al., 2019; Levy et al., 2020); however, they also constitute the main trigger of flash floods in arid and hyper-arid areas such as the Sinai Peninsula (Fig. 1), which hereafter is

referred to as SiP in this study (Ocakoglu *et al.,* 2002; David-Novak *et al.,* 2004; El-Magd *et al.* 2010; Farahat *et*
*al.* 2017; Gado, 2020).
The eastern Mediterranean is one of the main cyclogenetic regions of the Mediterranean basin (Krichak *et al.,*
1997) and globally (Ulbrich *et al.,* 2012; Neu *et al.,* 2013), which in many cases associated with precipitation
extremes (Flaounas *et al.,* 2014a, 2014b). As such, most of the heavy precipitation events in this region strongly
rely on the presence and frequency of the intense Mediterranean cyclones (Trigo *et al.,* 2002; Kotroni *et al.,* 2006;
Pfahl and Wernli 2012; Lionello *et al.,* 2016), accompanied by other precipitation producing-systems at synoptic-
scale, sometimes of tropical/sub-tropical origin (Krichak et al. 1997; Hochman *et al.,* 2020).
A multitude of observational-numerical-synoptic studies has been carried out in relation to the extreme
precipitation events over the eastern Mediterranean region to date, such as extreme rainfall analysis (e.g. Alpert *et*
*al.,* 2002; Ben David-Novak *et al.,* 2004; Kostopoulou and Jones, 2005; Ben-Zvi, 2009; Mathbout *et al.,* 2018),
trends in extreme precipitation (e.g. Yosef *et al.,* 2009; Shohami *et al.,* 2011; Ziv *et al.,* 2013; Ajjur and Riffi,
2020), satellite remote-sensing-based analysis of precipitation extremes (e.g. Gabella *et al.,* 2006; Mehta and
Yang, 2008; Nastos *et al.,* 2013; Yucel and Onen, 2014), numerical modelling and climate change projections of
heavy precipitations (e.g. Tous *et al.,* 2015; Romera *et al.,* 2016; Toros *et al.,* 2018; Zoccatelli *et al.,* 2020; Zittis
*et al.,* 2020), flash floods and water resources attributed to extreme rainfall events (e.g. Morin *et al.,* 2007; Samuels
*et al.,* 2009; Koutroulis and Tsanis, 2010; Tarolli *et al.,* 2012; Varlas *et al.,* 2018; Zoccatelli *et al.,* 2019; Spyrou
*et al.,* 2020; Rinat *et al.,* 2021), synoptic analysis of precipitation extremes and floods (e.g. Dayan *et al.,* 2001,
2015; Kahana *et al.,* 2002; Alpert *et al.,* 2004; Tsvieli and Zangvil, 2005; Peleg and Morin, 2012; Raveh-Rubin
and Wernli, 2015; Toreti *et al.,* 2016), and cyclogenesis and cyclone tracking (e.g. Alpert and Ziv, 1989; Alpert
and Shay-El, 1994; Flocas *et al.,* 2010; Flaounas *et al.,* 2014a, 2014b; Almazroui *et al.,* 2014; Zappa *et al.,* 2014;
Ziv *et al.,* 2015).
However, literature review of SiP reveals that very limited studies have been carried out so far mainly on the flash
floods associated with heavy rainfall events from the ground/satellite-based data analysis approach (e.g. Roushdi
et al., 2016; Dadamouny and Schnittler, 2016; Arnous and Omar, 2018; Morsy *et al.,* 2019; Baldi *et al.,* 2020) to
numerical model experiments (e.g. Cools et al., 2012; El Afandi *et al.,* 2013; Morad, 2016; Prama *et al.,* 2020;
Omran, 2020; El-Fakharany and Mansour, 2021). In such circumstances, Mohamed and El-Raey (2019) pointed
out that limited numbers of extreme precipitation events with high intensities and short durations that typically
result in flash floods allegedly are the only sources of the renewable water-resources in SiP. Therefore, it seems
necessary to understand, in the first place, the spatiotemporal distribution of extreme precipitation events across
SiP, and in the second place, to discover the corresponding synoptic-dynamical mechanisms responsible for the
occurrence of such events over the region. To our best of knowledge, no study attempted yet to quantify the
extreme precipitation characteristics (e.g. spatiotemporal variations, anomalies, frequencies and spatial patterns)
associated with the synoptic-regional atmospheric circulations and the cyclone tracking over SiP –and even not
over the eastern Mediterranean basin, as described and presented in this study. Yet, the wet and dry periods of SiP
have not been realized; it is of importance to the follow-up SiP's researches (e.g. assessing the rate of precipitation
recycling during the naturally dry-period of the year). Therefore, to bridge the above-mentioned research-gaps, in
this study, the following major research questions are addressed in particular during the SiP's wet and dry periods:
i.  how are the extreme precipitation climatologies spatiotemporally distributed across SiP?
ii.  which synoptic-scale systems are responsible for the occurrence of SiP's extreme precipitation events?
iii.  what are the major tracks of cyclones and their frequencies over the eastern Mediterranean region?
In this research, our data-analysis spans the period from 1st of January 2001 to 31st of December 2020. First, SiP's
wet and dry months are determined using a multi-statistical-approach developed in this study. Next, we use satellite
remote-sensing precipitation to quantify the spatiotemporal variations, anomaly, monthly regime, frequency and
spatial patterns of the extreme precipitation events, together with the computation of a set of extreme climate
indices, separately during the wet and dry periods. Then, the dominant synoptic atmospheric circulation patterns
and moisture condition corresponding to SiP's extreme precipitation events in wet and dry periods are explored
using the reanalysis data at multiple levels of the atmosphere. Finally, a daily-based frequency of the precipitation
producing-systems (cyclone tracking) are tracked and plotted over the region for the wet and dry periods.

## 2 Data and methods

### 2.1 The description of study area

The Sinai Peninsula (SiP, Lat: 27.6°N – 31.4°N, Lon: 32.2°E – 34.9°E) is located in the northeast of Egypt with an area of 61,000 km$^2$ (Fig. 1) covering about 6% of Egypt's area (Mohamed *et al.,* 2014; Badreldin and Goossens 2013). The region lies in an arid to hyper-arid belt of North Africa and belongs to the Saharan-Mediterranean climate classification (Dadamouny and Schnittler, 2016). Nevertheless, it is one of the coldest regions in Egypt due to its high altitudes and mountainous topography, where highest elevations are found toward the southern parts (e.g. Mount Catherine, the highest mountain in Egypt with an elevation of 2642 above ground level (AGL), see Fig. 1). Overall, SiP is characterized by a Mediterranean climate in the north and a semidesert/desert climate in the south (El-Sayed and Habib, 2008).

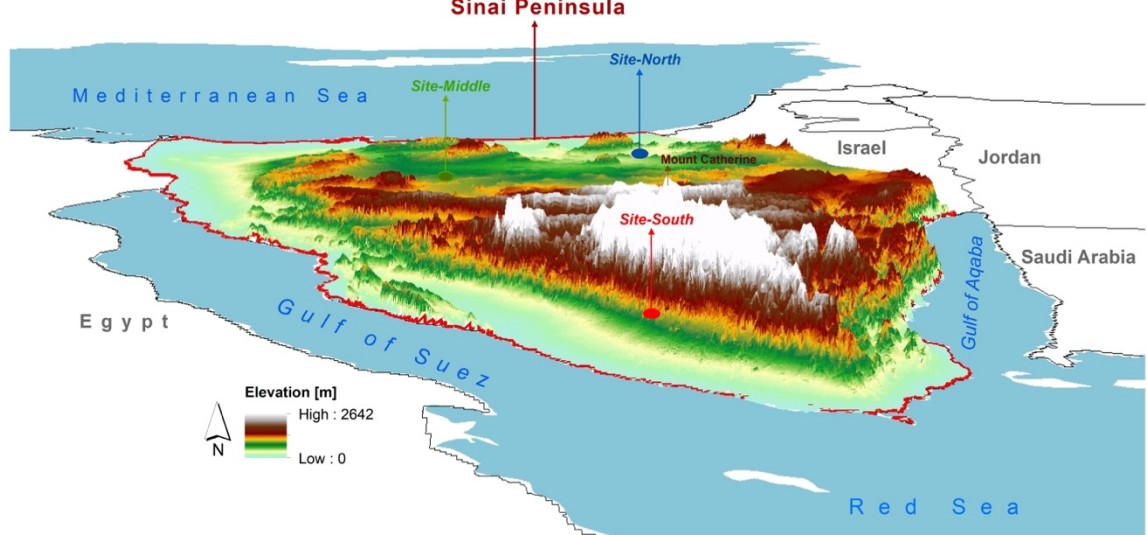

**Figure 1.** The location of Sinai Peninsula (SiP) in northeast of Egypt with the underling three-dimensional topography. Three selected sites in the north (Site-North: 30. 07°N, 33.09°E), middle (Site-Middle: 30.01°N, 33.50°E) and south (Site-South: 28.50°N, 33.70°E) of SiP shown here, used for the site-scale-based calculation of precipitation anomalies.

### 2.2 Datasets

2.2.1 Satellite Global Precipitation Measurement (GPM)

The Global Precipitation Measurement (GPM) is an international satellite mission to provide quasi-global precipitation estimates with a high temporal resolution (30min, daily and monthly) and spatial resolution (0.1°) through the Integrated Multi-satellitE Retrievals (IMERG) product. The GPM mission follows the Tropical Rainfall Measuring Mission (TRMM) program, aiming at improving the satellite-based precipitation observation capability. GPM-IMERG provides different rainfall estimates that are combined from active and passive instruments in the GPM constellation (https://gpm.nasa.gov/). Further detailed are given by Huffman *et al.,* (2014). The GPM data has been employed in several studies over the Mediterranean region (e.g., Retalis *et al.,* 2018; Petracca *et al.,* 2018; Caracciolo *et al.,* 2018; Cinzia Marra *et al.,* 2019; Hourngir *et al.,* 2021). In this study, we used the IMERG version 6 GPM-L3 final precipitation product (30min/daily) to estimate the extreme precipitation characteristics for a 20-year period (2001-2020) over SiP.

2.2.2 NCEP/NCAR and ERA5 reanalysis data

To investigate the synoptic/dynamical climatology associated with SiP's rainfall events, the required variables were obtained from National Centers for Environmental Prediction and National Center for Atmospheric Research (NCEP/NCAR) (Kalnay 1996) https://psl.noaa.gov/data/gridded/data.ncep.reanalysis.html and the fifth generation of the European Centre for Medium-Range Weather Forecasts (ERA5) (Hersbach et el., 2020) https://www.ecmwf.int/en/forecasts/datasets/reanalysis-datasets/era5. NCEP/NCAR and ERA5 provide the reanalysis datasets with multiple time-steps at the surface and pressure levels of the atmosphere since 1948 and

1979 with a global 2.5° × 2.5° and 0.25° × 0.25° horizontal grid, respectively. In the literature, these datasets have
been used over the Mediterranean region in several studies especially with regard to the synoptic analysis of
precipitation, blocking systems, storm and cyclone tracking (e.g. Krichak *et al.,* 2002; Trigo *et al.,* 2004; Tolika
*et al.,* 2006; Trigo, 2006; Lois, 2009; Barkhordarian *et al.,* 2013; Almazroui and Awad, 2016; Almazroui *et al.,*
2014, 2017; Varlas *et al.,* 2018; Kotsias et al., 2020). First, in this research, NCEP/NCAR data was used to study
the pressure fields due to its coarser resolution, as it is believed that large-scale pressure systems such as cyclonic
-and anticyclonic patterns could be better represented in a coarse resolution especially at lower atmospheric levels
over the complex environments. Second, ERA5 data was used to quantify the moisture condition and wind streams
structure/profile related to the wet -and dry periods at a finer resolution. The following reanalysis meteorological
dataset -or derived variables at multi-levels were employed: NCEP/NCAR (daily, 250km grid), sea level pressure
SLP (hPa), geopotential height HGT (m), relative vorticity RV ($10^{-5}$ $S^{-1}$), zonal (U) and meridional (V) wind-
components (m $s^{-1}$), and vertical velocity (omega: Pa $s^{-1}$); and ERA5 (daily, 25km grid), relative humidity RH (%),
and U and V wind-components (m $s^{-1}$).

**2.3 Data analysis approach**
2.3.1 Determining SiP's wet and dry periods
In this research, months with lowest (-or no rainfall) and highest amounts/frequencies of the precipitation events
are determined throughout the year in SiP. This is important, as in the follow-up SiP's researches it is aimed to
assess the regreening impacts on local hydrometeorological processes such as precipitation recycling in SiP under
a vegetated-surface scenario during a naturally dry period of the year. Thus, we developed a multi-statistical-
approach to split the wet -and dry months of the year for the period 2001-2020. This is achieved via a combination
of the results obtained from three statistical measures: i) monthly 90[th] percentile (Fig. 3a), ii) frequency occurrence
of precipitation with a threshold of ≥10 mm/day (Fig. 3b) – after examining other thresholds of ≥5 mm/day and
≥20 mm/day (see Figs. S1 and S2), and iii) monthly rainfall standard deviations (Fig. 3c). These methods were
calculated using a set of statistical functions described in the follow-up subsection (see Table 1). Therefore, using
the approach developed in this study, wet months are determined from October to March defined as wet-period,
and dry months from April to September defined as dry period in SiP.

2.3.2 Estimate of the extreme precipitation indices and statistical values
The spatiotemporal analysis and statistical measures on the satellite GPM-based daily precipitation timeseries was
carried out for the entire SiP region. For this, a set of climate functions/indices (see Table 1 for the details) was
computed for the period of 2001-2020 using the Climate Data Operator (CDO) (Schulzweida, 2020), developed
in Max-Planck-Institute for Meteorology (https://code.mpimet.mpg.de/projects/cdo).

**Table 1.** CDO functions and climate indices used in this study (Schulzweida, 2020).

| Index | Descriptive name | Definition | Units |
|---|---|---|---|
| *monsum* | Monthly sum | For every adjacent sequence $t\_1$, ..., $t\_n$ of time steps of the same month it is: o(t, x) = sum{i(t', x), $t_1$ < t' ≤ $t_n$} | mm |
| *yearsum* | Yearly sum | For every adjacent sequence $t\_1$, ..., $t\_n$ of time steps of the same year it is: o(t, x) = sum{i(t', x), $t_1$ < t' ≤ $t_n$} | mm |
| *eca_pd* | Precipitation days index per time period | Generic ECA operator with daily precipitation sum ≥5 mm. | days |
| *eca_r10mm* | Heavy precipitation days index per time period | Specific ECA operator with daily precipitation sum ≥10 mm | days |
| *eca_r20mm* | Very heavy precipitation days index per time period | Specific ECA operator with daily precipitation sum ≥20 mm | days |
| *eca_cdd* | Consecutive dry days index per time period | Maximum number of dry days with daily precipitation sum ≥1 mm | days |
| *eca_rr1* | Wet days index per time period | Number of wet days with daily precipitation sum ≥1 mm | days |
| *eca_sdii* | Simple daily intensity index per time period | Average precipitation on wet days with daily precipitation sum ≥1 mm | mm |

| | | | |
|---|---|---|---|
| *timstd* | Time standard deviation | Total monthly precipitation ≥1 mm | mm |
| *monpctl,90* | Monthly 90ᵗʰ percentile | Total monthly precipitation ≥1 mm | mm |

### 2.3.3 Calculation of the precipitation spatiotemporal variations

The spatiotemporal patterns of the daily precipitation climatology (annual and biannual) over the period of 2001-2020 in SiP were analyzed using the Empirical Orthogonal Function (EOF) analysis. According to Dawson (2016), the main aim of EOF analysis is to reduce the dimensionality of a spatial-temporal dataset by transforming it to a new basis in terms of variance. This transformation turns the input spatial-temporal dataset into a set of maps representing patterns of variance, and a timeseries for each map that determines the contribution of that map to the original dataset at each timestep. Thus, the spatial patterns are the EOFs, and are considered as basis functions in terms of variance. The associated timeseries are the principal components (PCs) and are the temporal coefficients of the EOFs. In this study, we used a Python-based *eofs package* (Dawson, 2016) to perform the EOF analysis. Furthermore, the trends of the annual and seasonal changes in the precipitation events were also estimated for the three selected sites across SiP (see Fig. 1 for the locations) using anomaly-based analysis. The climatology mean precipitation values and spatial distribution were the two main criteria for the selection of the sites. In this way, each chosen site is representative for its surrounding area in terms of both the precipitation magnitude and spatial patterns. Thus, the selected sites in north, south and middle parts indicate the max, min and average amounts of precipitation received across SiP, respectively over a 20-year time-period. For this analysis, the precipitation anomalies (annual and seasonal) are calculated in three steps: i) calculating the climatology mean of the data, ii) subtracting the mean value from each year/season values, and ii) drawing the trend of slopes using the least squares method. Here, winter includes DJF months (Dec, Jan and Feb) and autumn includes SON months (Sep, Oct and Nov). The anomalies for spring and summer periods were found to be close to zero, and therefore excluded. It is noted that, we also performed the 95% and 99% bootstrapped confidence intervals for the Mean and Median values of the original dataset (seasonal and annual) for the selected sites. The results are given in table S1.

### 2.3.4 Synoptic analysis

To explore the climatology of the synoptic, dynamics and moisture condition at multiple level of the atmosphere responsible for the occurrence of the (extreme) precipitation events over SiP, the reanalysis dataset obtained from NCEP/NCAR and ERA5 was investigated. In the first place, the wet-period and dry-period were determined as explained earlier (see Sect. 2.3.1). In the second place, using the satellite-reanalysis variables (see Sect. 2.2.), the dominant synoptic features, dynamical circulation patterns and moisture condition accompanied by the spatial correlations between SiP's rainfall and key meteorological variables were computed and analyzed for the wet and dry periods for the climatology period of 2001-2020.

### 2.3.5 Cyclone tracking

In line with the synoptic analysis, the daily trajectories of the rainy-systems precipitated over SiP were tracked and plotted for the wet -and dry periods using a manual approach developed in this study. In our approach, we merely aimed to detect and track cyclones precipitated with ≥10mm in SiP. This is however challenging for an automated algorithm to detect a low system (sometimes with multiple centers, cyclones) that may -or not has generated a rainfall with a given threshold over a given domain. Yet, its performance is not totally error-free in particular over the heterogeneous regions with complex atmospheric PBL like the Mediterranean region (e.g., Raible *et al.,* 2007; Flaounas *et al.,* 2014c; Prantl *et al.,* 2021). Our manual-based cyclone-tracking approach developed in this research consists of three major steps as follows:
*i) first*, a set of daily total precipitation patterns over SiP was produced using GPM data separately for the wet -and dry periods; by doing so, a total number of 156 events (out of 7305 days) were identified, for which precipitated ≥10mm over SiP. Accordingly, the synoptic-scale daily composites of SLP, and 850-hPa RV and streamflow were produced using the reanalysis dataset for the entire study-period (2001-2020, 7305 days). Here, the 850-hPa relative vorticity -and streamflow were used along with SLP to better identify the lows (Flaounas *et al.,* 2014c).
*ii) second*, to identify the cyclogenesis/lysis of the selected events, the composite maps of SLP, RV and streamflow for several days before -and after SiP's precipitation events were monitored and tracked carefully. Every daily movement (X-Y coordinates) of the corresponding cyclone was recorded from the beginning where the low system

was born (cyclogenesis) until it was disappeared (cyclolysis). This process was carried out one-by-one for all 156
cases with rainfall ≥10mm. All the events were classified into five categories based on the rainfall magnitude as
follows: category 1 (10-20mm), category 2 (21-30mm), category 3 (31-40mm), category 4 (41-50mm) and
category 5 (>51mm). *iii) third*, finally the cyclone tracking charts for the wet -and dry periods were separately
produced using the information obtained from the former steps.

## 3 Results
**3.1 Climatology analysis of the precipitation characteristics**
3.1.1 The precipitation spatial patterns and extreme indices
The spatial precipitation patterns in terms of the climatology average, the rainiest month, and the wettest day for
the period of 2001-2020 in SiP are illustrated in Figure 2a-c, respectively. The climatology map of precipitation
markedly demonstrate that northeast and southwest parts of SiP receive the highest (ranged between 100 and 150
mm/year) and the lowest (ranged between 20 and 30 mm/year) amounts of annual rainfall, respectively (Fig. 2a).
This implies that the precipitation unevenly distributed over SiP. However, most parts of the region receive not as
high as 40 mm/year, except for the northern areas close to the Mediterranean Sea. With respect to the occurrence
of the precipitation extremes, we discovered that the rainiest month (out of 240 months) was in March 2020 (Fig.
2b) with a wide range of rainfall values from 15 to 30 mm/month in the south and from 50 to 70 mm/month in the
north. Interestingly, the wettest day (out of 7305 days) has been also occurred in the same month/year, which is
March 12, 2020 (Fig. 2c), thus it is not surprising to see an analogous spatial pattern when compared to the rainiest
month but with less magnitude.

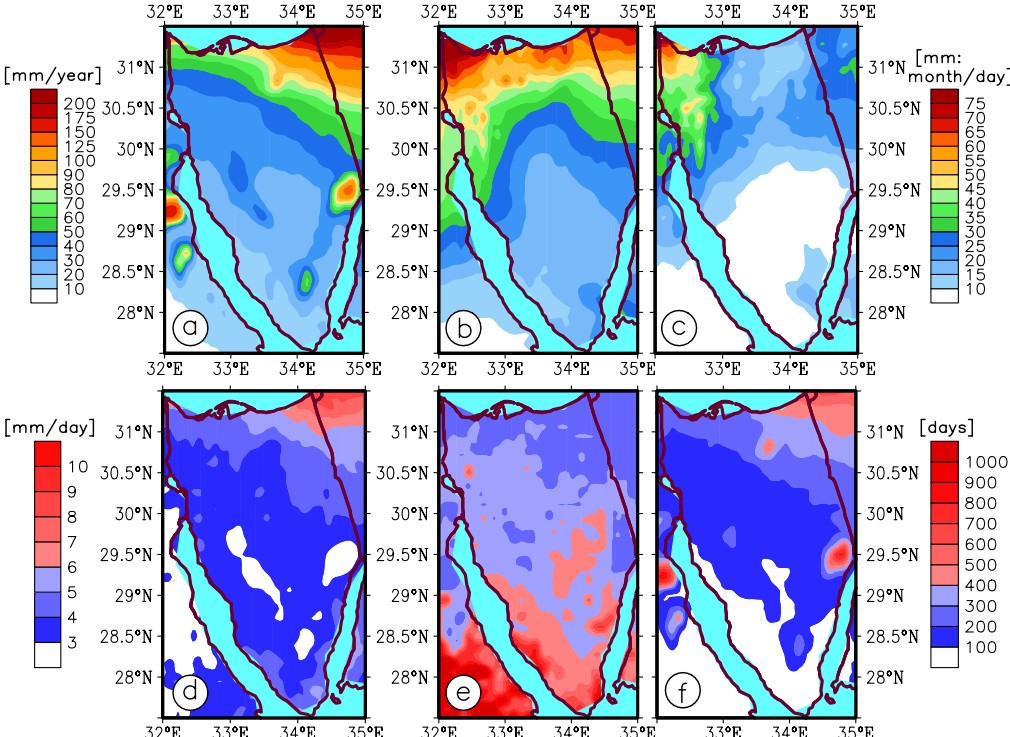

**Figure 2.** The precipitation spatial patterns and extreme indices: a) climatology map of mean annual precipitation (2001-2020);
b) the wettest month i.e. March 2020 (out of 240 months), c) the wettest day i.e. March 12, 2020 (out of 7305 days); as well as
extreme daily precipitation indices with a threshold of ≥1mm/day: d) simple daily intensity index (SDII), e) consecutive dry
days (CDD) and f) wet days index (RR1) for the period of 2001-2020 over the Sinai Peninsula (SiP).

Additionally, we also identified the twelve rainiest months out of 240 months (see Fig. S3) and the twelve wettest
days out of 7305 days (see Fig. S4). It was found that 9 out of 12 extreme month/day cases occurred in the winter
season (Jan, Feb and Mar) with the highest frequency occurrence in January (5 cases); while only 3 out of 12 cases
took place in autumn (Oct and Dec). Further, we plotted monthly precipitation climatologies (2001-2020) together

with ranks of 12 months (out of 240) with the highest amount of rainfall received in SiP (Fig. S5). The most extreme precipitation event occurred in March 2020 over the past two decades, followed by February 2019 and January 2013. The severest storm recorded during 11-13 March, and the peak rainfall hours (>30mm) occurred in the afternoon of the March 12, 2020, as shown by onset and termination of the most powerful rainy-system in hourly intervals of the subplot in Figure S5c. It may worth mentioning that the exceptional storm event of 11-13 March 2020 over SiP is comprehensively investigated via data-analysis and simulation-experiment approach in a follow-up research. Overall, in almost all the precipitation cases either in climatologies or extremes, a similar spatial precipitation pattern was captured, meaning that the maxima were recorded in the north and the minima in south of SiP.

As shown in Fig. 2d-f, the dryness and wetness conditions across SiP were also explored by computing the simple daily intensity index (SDII), number of consecutive dry days (CDD) and number of wet days (RR1). It can be clearly seen that the highest SDII is observed in the northeast with an intensity of ≥6 mm/day. Interestingly, the lowest SDII is not seen in the south (even though the minimum precipitation magnitude and frequency is located there – see Fig. 2a), but in central parts of SiP with ≤3 mm/day (Fig. 2d). CDD is remarkable in the south with ≥500 out of 7305 days, indicating that these areas receive less than 1mm rainfall for a long period; however, it is gradually decreasing northward with ≤300 days (Fig. 2e). Unlike CDD, it is not surprising to observe that RR1 is the lowest in the south (≤100 days) and innermost parts (≤200 days), but rapidly increases towards the northeast of the region (≥350 days), as shown in Figure 2f. These results are in good agreement with the above-mentioned findings over SiP.

3.1.2 The wet and dry period's monthly precipitation patterns: a multi-statistical analysis

*i) percentile-approach* (Fig. 3a): monthly 90[th] percentile of precipitation reveals that percentiles ≥10 mm/month are merely observed from October to March (wet period); while for the period from April to September (dry period) very low -or no rainfall are realized, suggesting a prolonged naturally dry-period in SiP. Further, temporally, the winter months comparatively receive higher values (of extreme rainfall with 90[th] percentile), when compared to the autumn months (with <25 mm/month) during the wet period. Spatially, percentile maxima >50 mm/month are only seen in SiP's northeast across the year.

*ii) frequency-approach* (Fig. 3b): frequency occurrence of heavy precipitation ≥10 mm/day at a monthly basis are almost limited to the wet period of winter months (ranging from 1 to 40 days/month) and autumn months (ranging from 1 to 25 days/month). It is noteworthy that, the frequency occurrence of rainfall ≥ 20 mm/day reduced by half in comparison with ≥10 mm/day occurring mostly in the late autumn and early winter episodes, and yet limited only to a small part of the northeastern SiP (see Fig. S2). Further, the annual frequency occurrence of the SiP's rainfall extremes shows that the highest and lowest frequencies with a threshold of ≥5 mm/day are occurred in the north (ranged between 100 and 250) and south of SiP (ranged between 20 and 40), respectively. Higher thresholds of ≥10 mm/day and ≥20 mm/day however follow the same spatial pattern as to threshold of ≥5 mm/day across the region, but with lower frequencies (see Fig. S1). Nevertheless, distribution of the frequencies, regardless of their thresholds, are in very good agreement with the spatial pattern of precipitation climatology (Fig. 2a).

*iii) standard deviation-approach* (Fig. 3c): magnitude of precipitation variability, as given by standard deviation SD, reveals a similar spatial pattern as to the percentile and frequency patterns across SiP region. This implies that the northern SiP shows the highest variability with at least 10 mm/month during wet period, while the reverse is true for the dry period with almost no rainfall except for April and May with the lowest standard deviation (<7 mm/month). Also, variability is largest in March over the northern SiP from a spatial view.

Overall, the results obtained from the three statistics used are quite concordant and compatible with respect to the SiP's spatial precipitation variability at a monthly basis; and suggest that the (extreme) precipitation events are inherently limited to the wet period from October to March, whereas months from April to September receive very low -or no rainfall at all during the dry period (Fig. 3).

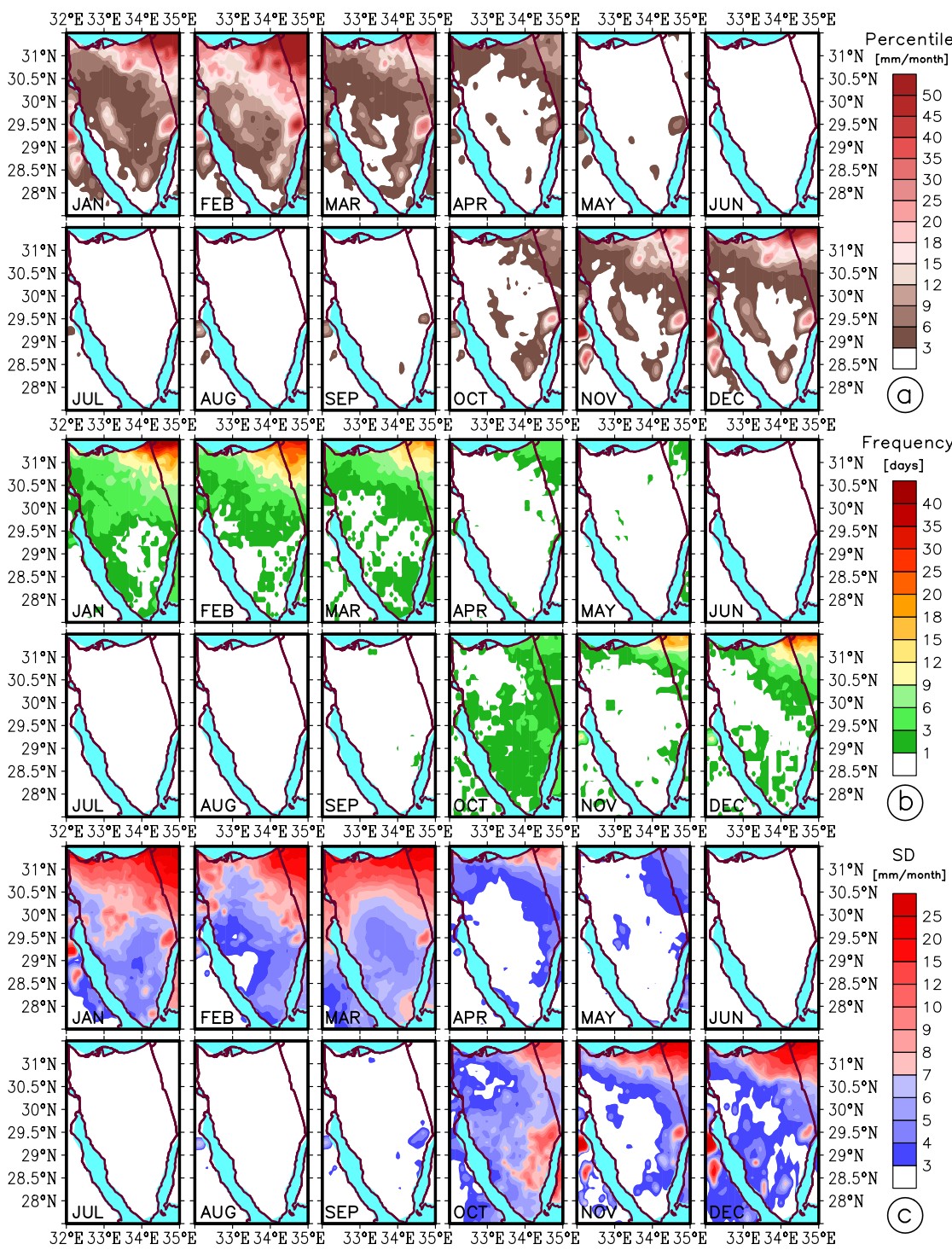

302

**Figure 3.** A multi-statistical analysis of the precipitation at a monthly basis: a) the 90th percentile of rainfall climatology, b) frequency occurrence of rainfall events with a threshold of ≥10 mm/day, and c) grid-based standard deviation estimates of rainfall for the period of 2001-2020 over the Sinai Peninsula (SiP).

306

### 3.1.3 Spatiotemporal variations of the precipitation: EOF-based analysis

To investigate the patterns of precipitation variabilities in time and space in SiP, EOF analysis was performed on the monthly precipitation dataset in the annual scale (Figure 4). The first two leading EOFs account for 60% and 11% of the variance, respectively. The EOF1 spatial pattern is entirely in negative mode SiP-wide, indicating a below-average-rainfall condition (drier trend) especially in northern SiP (Fig. 4a). Correspondingly, PC1 time-series indicates a dominant negative temporal variability of the EOF1 for the entire period (Fig. 4b). Conversely,

it is seen that EOF2 values are mostly in positive mode showing an above-average-rainfall condition (wetter trend)
in most parts in particular in the southern SiP (Fig. 4c), and the positive temporal variability of the EOF2 is mostly
seen in the recent years, as shown in the PC2 timeseries (Fig. 4d). However, the northern SiP remains in negative
mode suggesting a severe-decreasing trend in the annual precipitation rate when it is combined with the EOF1,
which 60% of variance explained.
Besides annual analysis, seasonal spatiotemporal variabilities of the EOF-patterns were also performed separately
for wet-period (Fig. S6) and dry-period (Fig. S7). We found that, annual EOFs/PCs strongly resemble the seasonal
EOFs spatial patterns and PCs temporal variabilities in the SiP's wet-period. This implies that, both wet-period
and annual patterns capture a decreasing trend in the north and an insignificant increasing trend in the south of
SiP. It is also noted that, grid-based spatiotemporal variations obtained by the EOF-analysis are in good agreement
with the site-scale anomaly-based temporal changes in the annual -and seasonal precipitation trends observed in
the selected sites across SiP region (see Fig. S8 for the details).

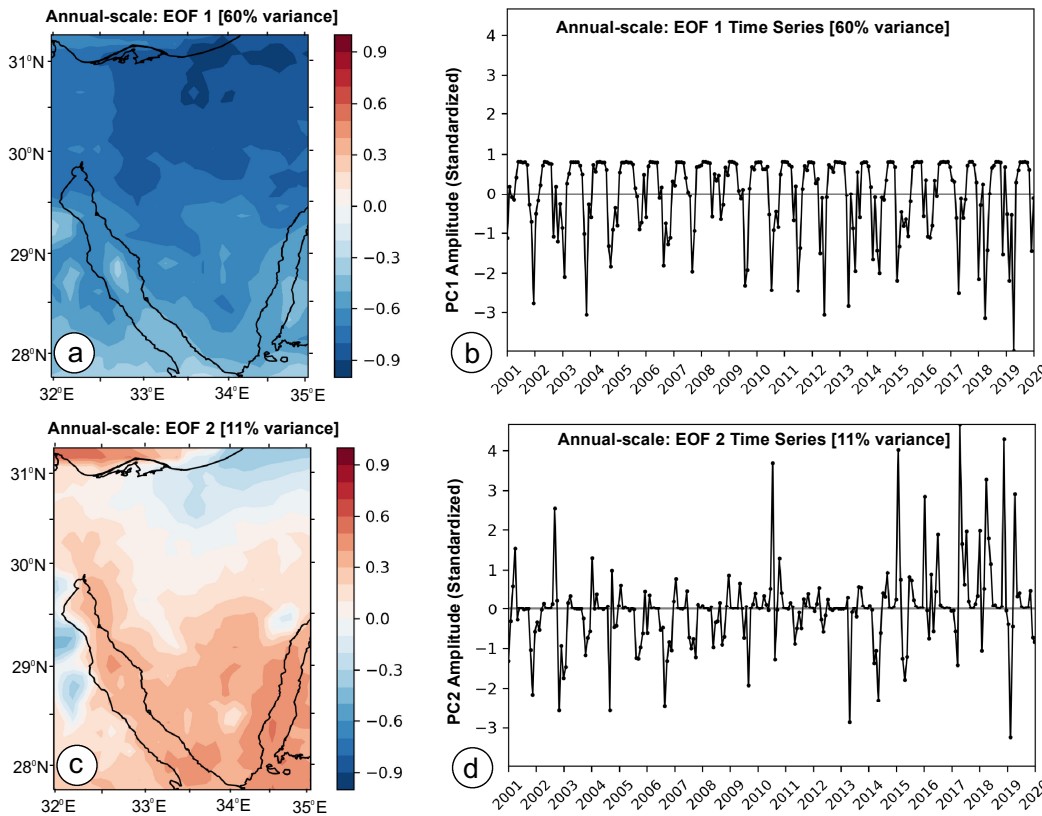


**Figure 4.** The two leading EOF spatial patterns (a and c) and associated timeseries (b and d) of the monthly mean precipitation
dataset (at annual scale) for the period of 2001-2020 (240 months) in the Sinai Peninsula (SiP). The values of EOFs (a and c)
are expressed as correlation coefficients.

3.1.4 Monthly regime of the precipitation climatology
Figure 5 represents the precipitation regime climatology with respect to the ratios and standard deviation estimates
at a monthly basis over SiP. High (>20%) ratios of monthly precipitation over annual precipitation are estimated
in the winter months of January and February, mostly found in the mid-to-north of SiP. March indicates some
patches of high ratios in south and northwest also, as shown in Fig. 5a. However, the period of April to September
(colored in black in the legend) receives less than 20% ratio of the annual precipitation. This implies that spring
and summer months experience longer dry weather periods than winter season. Considering the autumn months,
the areas with 20% ratio of annual precipitation remain largely out of SiP domain, expect for a few mini-patches.
Therefore, winter is the rainiest season throughout SiP. Besides, the monthly SD estimates (Fig. 5h) also follow a
pattern similar to the ratios across the year. This means that, temporally winter (summer) months hold the highest
(lowest) variation values, and spatially northern (southern) SiP possesses the highest (lowest) values with a max
value of 17.0 mm/month estimated in March in northeast of SiP. It is also noted that, the full ratios of monthly to
annual precipitation for individual months of the year are illustrated in Figure S9, as well as the full grid-based SD
estimates for the entire SiP in a monthly basis represented in Fig. 3c, which could provide further details on SiP's
precipitation regime climatology.
Furthermore, to compare the precipitation monthly ratios across SiP, the bar charts for the given sites covering the
whole SiP were plotted (Figs. 5b-5g). The highest and lowest ratios are found in winter and summer months,
respectively. However, by a closer look it becomes clear that chosen sites do vary in terms of magnitude and trends
in the monthly precipitation ratios. For instance, in most sites the highest monthly ratio is observed in February
(>18%), except for the sites located in SiP southwest (which, is January with >19% – Fig.5d) and southeast (which,
is March with >17% - Fig.5g). Likewise, an inconsistent seasonal trend is also remarkable for the autumn months,
meaning that the northern sites indicate a positive trend from the late summer to the end of autumn (Figs. 5a, 5b
and 5e). The southern sites, however, represent a contrasting pattern with respect to monthly rainfall regime (Figs.
5d, 5f and 5g).

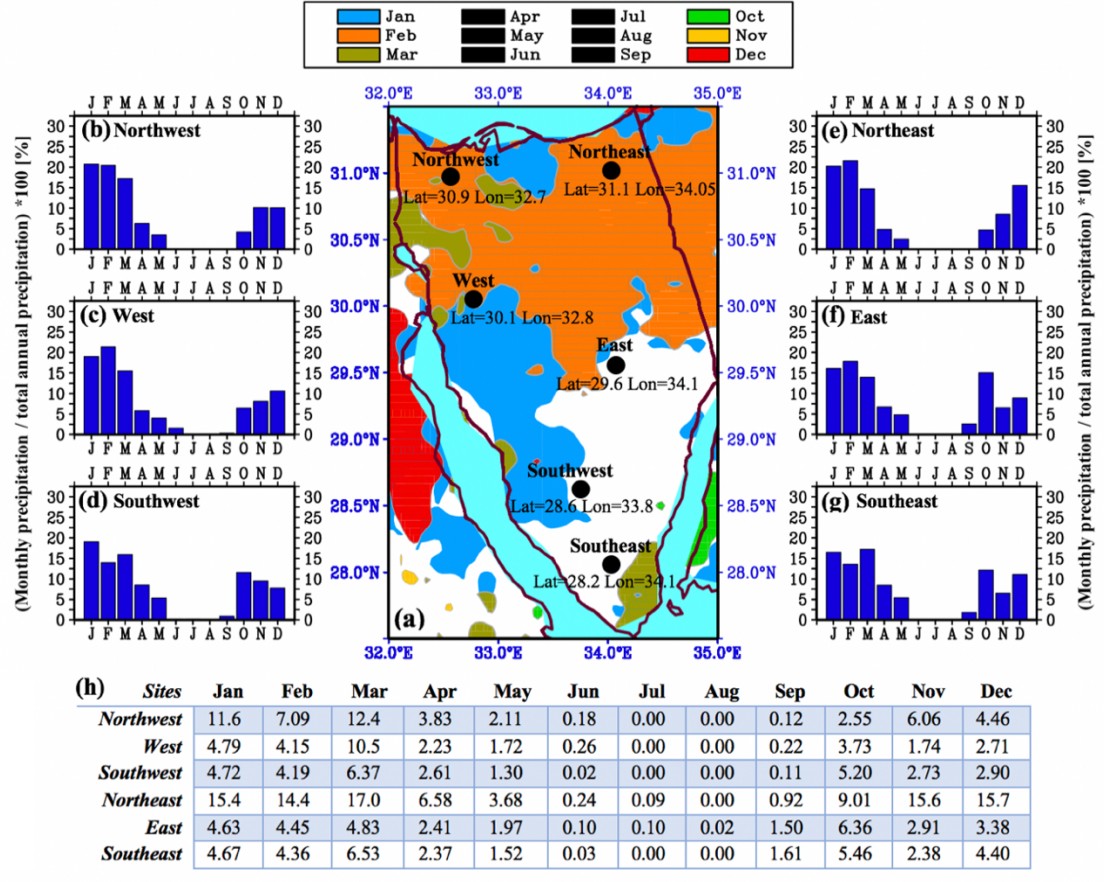

| (h) Sites | Jan | Feb | Mar | Apr | May | Jun | Jul | Aug | Sep | Oct | Nov | Dec |
|-----------|-----|-----|-----|-----|-----|-----|-----|-----|-----|-----|-----|-----|
| Northwest | 11.6 | 7.09 | 12.4 | 3.83 | 2.11 | 0.18 | 0.00 | 0.00 | 0.12 | 2.55 | 6.06 | 4.46 |
| West | 4.79 | 4.15 | 10.5 | 2.23 | 1.72 | 0.26 | 0.00 | 0.00 | 0.22 | 3.73 | 1.74 | 2.71 |
| Southwest | 4.72 | 4.19 | 6.37 | 2.61 | 1.30 | 0.02 | 0.00 | 0.00 | 0.11 | 5.20 | 2.73 | 2.90 |
| Northeast | 15.4 | 14.4 | 17.0 | 6.58 | 3.68 | 0.24 | 0.09 | 0.00 | 0.92 | 9.01 | 15.6 | 15.7 |
| East | 4.63 | 4.45 | 4.83 | 2.41 | 1.97 | 0.10 | 0.10 | 0.02 | 1.50 | 6.36 | 2.91 | 3.38 |
| Southeast | 4.67 | 4.36 | 6.53 | 2.37 | 1.52 | 0.03 | 0.00 | 0.00 | 1.61 | 5.46 | 2.38 | 4.40 |

**Figure 5.** Monthly precipitation regime: (*a*) ratio of monthly sum precipitation to the annual total precipitation (%), where only
ratios >20% are plotted for each month; panels (*b-g*) indicate the monthly ratios (January to December) for the selected sites;
and panel (*h*) represents the standard deviation estimates (mm/month) in a monthly basis for each site shown in panel (*a*) across
the Sinai Peninsula (SiP) for the climatology period of 2001-2020. It is also noted that in the panel *a*, monthly ratios from April
to September (colored in black in the legend) are below 20%, thus not plotted here, but full ratios (%) are illustrated in Fig. S9
in a monthly basis. In addition to the panel (*h*), full grid-based standard deviation estimate for the entire SiP in a monthly basis
is also represented in Fig. 3c.
## 3.2 Synoptic analysis of the wet and dry periods
Spatial distribution of the monthly mean precipitation amounts and magnitudes indicated a remarkable difference
between the wet period (ranged 5-70 mm/month) and the dry period (ranged 1-3 mm/month) for the climatology
period of 2001-2020. However, despite a large dissimilarity in precipitation values of the wet and dry
periods, their spatial pattern climatologies largely resemble (see Fig. S10). This implies that, amount of rainfall in
both periods are notably increased from the southern parts towards northeast of SiP. In the follow-up subsections,
therefore, the large/regional-scale atmospheric systems corresponding for the occurrence of precipitation events
during the wet and dry periods of SiP are explored from a synoptic/dynamic -and moisture condition perspective.

3.2.1 Synoptic patterns and atmospheric circulation structure
Figure 6 represents the climatology of the synoptic patterns and cyclogenesis at suface and 500-hPa atmospheric
levels during the wet and dry periods over the Mediterranean basin including SiP (marked by a red box). In the
wet period at surface level (Fig. 6a), two major sources of strong cyclonic activities (cyclogenesis) are observed
over the Mediterranean westen part (at the lee of Alps Mountains over Gulf of Genoa) and eastern part (at the lee
of Taurus Mountains over Cyprus) – see Fig. 12 for the locations. These areas are found by the closed SLP contours
along with strong positive vorticity in the west and east parts of the Mediterranean Sea, respectively. Cyprus low
alegedly is responsible for the occurrane of majority of rainfall events over the eastern Mediterranean including
SiP. For instance, about 80% of the rainfall in the cold period of Israel are associated with the Cyprus cyclone
systems, as pointed out by Saaroni *et al.,* (2010). In wet period, the Red Sea trough as a lower-level system, is
another significant synoptic system that influencing the eastern Mediterranean region, but mostly in the autumn
(Ziv *et al.,* 2021). As shown in Fig. 6a, this trough is developed as a result of the coexistence of the eastern African
cyclone namely Sudan's Low and Saudi Arabia's anticyclone. Its high impact on the eastern Mediterranean area
is depending on the position of the Red Sea trough axis, that is, the eastern position, as pointed out by e.g. Saaroni
*et al.* (1998), and Tsvieli and Zangvil, (2005). However, the impact of the Red Sea trough on SiP's precipitation
is limited compared to northeastern parts of the Mediterranean basin, mostly due to the geographical location of
SiP. In line with lower levels, the pressure pattern at 500-hPa level shows a synoptic-scale trough (of the persistent
low center) with high positive vorticity providing a suitable condition for occurrence of rainfall events over the
Mediterranean region extending towards the Middle-East areas (Fig. 6b).

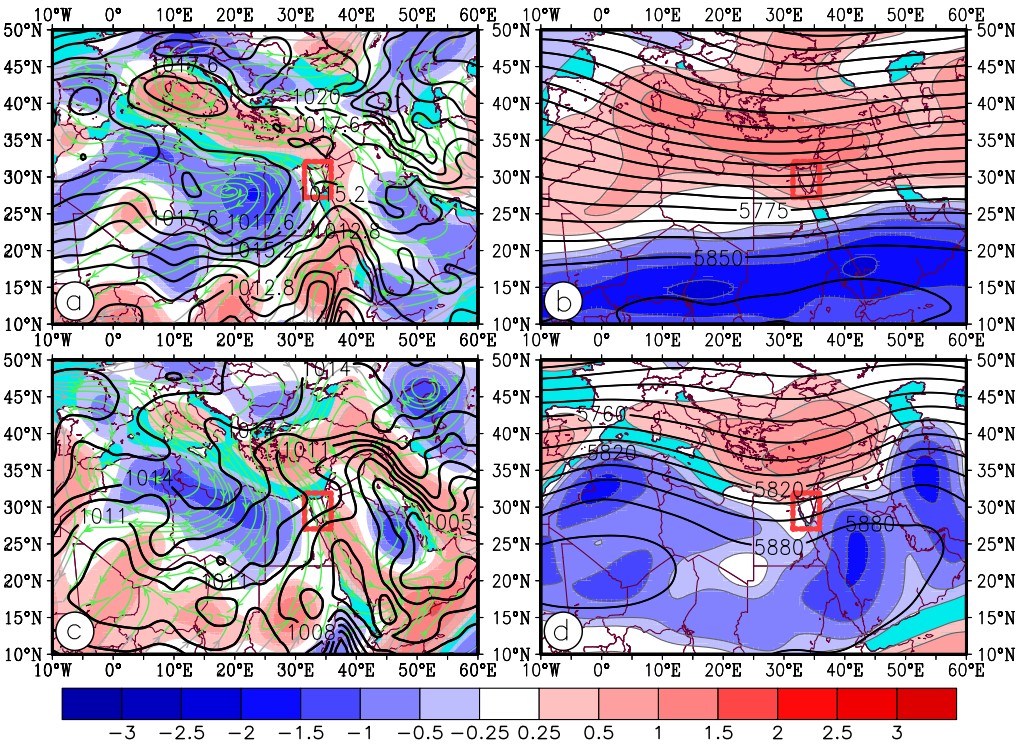


**Figure 6.** Climatology synoptic condition during the wet-period from October to March (a and b) and dry-period from April
to September (c and d) during the period of 2001-2020 over the Sinai Peninsula (SiP) (red box in each panel); a) composites
of sea level pressure (*black contours*, hPa), 925-hPa relative vorticity (*shading*, $10^{-5}$ $S^{-1}$) and streamflow (green streamline); b)
500-hPa composite of geopotential height (*isolines,* m) and relative vorticity (*shading,* $10^{-5}$ $S^{-1}$); c and d same as in *a* and *b*
panels respectively, but for the dry period.

In contarst to the wet period, surface level pattern of the dry period differs strongly over the region (Fig. 6c). In
the dry period, hardly ever cyclones are produced in the western Mediterranean as dominated by the high-pressure
systems extending from the north Atlantic Ocean and north of Africa. Limited low-pressure systems however are
typically developed over the eastern Mediterranean. This is due to the formation of a trough extending from the
Persian Gulf (which, developes as the result of the topographic impact of Zagros Mountains in western Iran) via
Taurus Mountains in the southern Turkey into the eastern Mediterranean basin (see Fig. 12 for the locations). The
SiP region locating in the southeastern Mediterranean basin, as shown in Fig. 6c, is highly influenced by the ridge
of the north Africa so-called Azores anticyclone, rather than the Persian Trough that impacting mostly the
northeastern Mediterranean. Thus, at midlevel of 500-hPa geopotential height, the eastern Mediterranean is mostly
subjected to persistent air subsidence, and only a limited trough is formed with relatively high positive vorticity
over the eastern Mediterranean (Fig. 6d). This results in preventing rainfall to a large extent over the region during
the dry period. Therefore, SiP recieves much less amount of precipitation in terms of magnitude and frequency,
compared to those received over northeastern parts (such as Israel) of the Mediterranean basin. These results are
in good agreement with the findings reported by Alpert *et al.* (1990) and Saaroni and Ziv (2000).

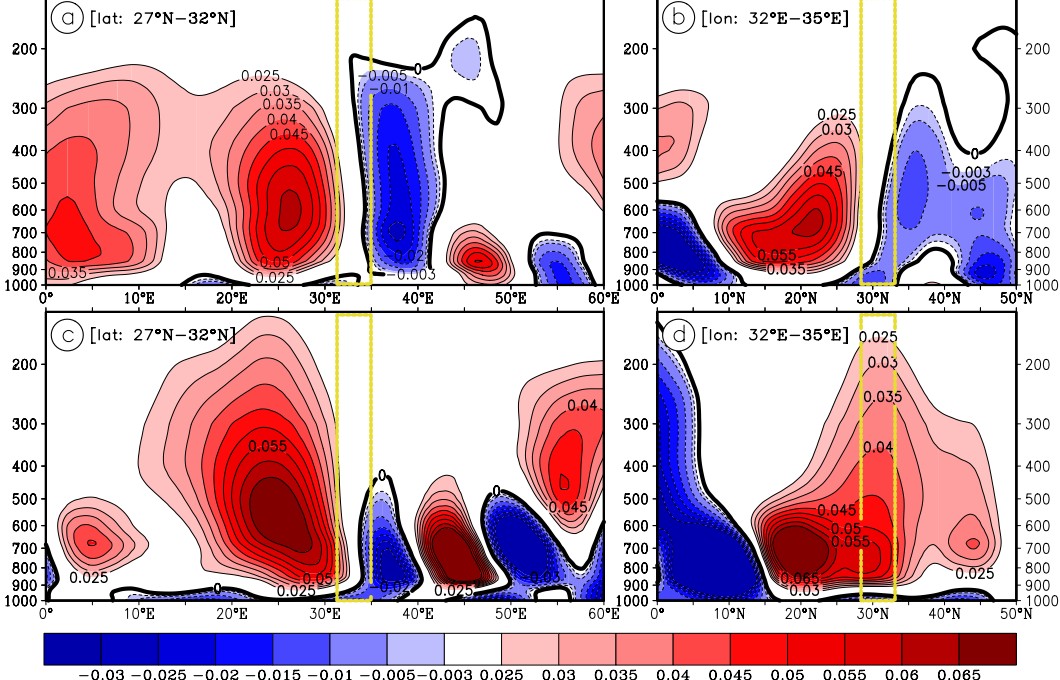

**Figure 7.** Vertical velocity cross-section (omega: Pa s$^{-1}$) for: the wet period of October to March (a and b), and dry period of
April to September (c and d) over the period of 2001-2020. Omega values averaged for the latitudes of 27°N-32°N across the
longitude (a and c panels), and for the longitudes of 32°E-35°E across the latitude (b and d panels). Yellow box in the panels
indicates the location of Sinai Peninsula (SiP).

Besides the synoptic pressure-systems described above, the veritical velocity motions (omega) could further reveal
discrepancies between the wet and dry periods, from a dynamical perspective. Increase in the synoptic precipitatin
events over the wet period is inevitably attributed to the existance and duration of strong rising parcls of air and
upward vertical streams over SiP and in the nearby regions. The omega cross-section along the longitude (Fig. 7a)
represents a maximum core with negative value of -0.03 Pa s$^{-1}$ occurs at 800-700 hPa levels (at above 36°E)
extending up to 250 hPa. It also indicates that, unlike to western parts, eastern parts of SiP experiencing a relatively
strong rising condition at multiple levels of the atmosphere during the wet period. A similar pattern analogous to
longitude cross-section is also observed along the latitude (Fig. 7b). This means that, the maximum core of vertical
velocity with the value of -0.006 Pa s$^{-1}$ is seen towards the northeast of the Siani (at below 35°N) in particular at
higher levels. However, when it comes to dry period, a much weaker negative omega is observed, mostly limited
to lower levels of the atmosphere along the longitude (Fig. 7c), and it is alegedly positive (sinking) in particular
on southern parts of the Sinia along the latitude (Fig. 7d). In such circumstances, the rising of air is strictly
restricted. This (Fig. 7) therefore further clarifies, among others, why the northeast parts of SiP receive higher
(intense) amount of precipitation compared to rest of SiP, that is, partially due to the stronger vertical velocity
motions in both the dry and especially wet periods.

### 3.2.2 Moisture transport and wind structure

Figure 8 illustrates the climatology of moisture condition and wind patterns separately for the wet -and dry periods in SiP (red box) -and in the nearby areas. Overall, a remarkable difference is observed with regard to the moisture availability during the wet -and dry periods in the region especially over SiP. During wet period, the prevailing westerlies at 850-hPa (which, is typically considered as condensation level) over the Mediterranean Sea along with the presence of an anticyclonic circulation pattern over north of Africa resulted in transferring abundant moisture (on average 50-70%) to the eastern parts of the Mediterranean basin including SiP (Fig. 8a). Also, the vertical cross-sections of moisture content and wind profile at pressure levels indicate that majority of the moisture needed for condensation is found at lower levels (950-850-hPa) over the region (Fig. 8b). The above-mentioned moisture and wind patterns however largely differ (RH reduced on average 25-45%) during the dry period at 850-hPa level (Fig. 8c) and pressure levels (Fig. 8d). This could be as the result of displacement of northern Africa's high-pressure center towards the higher latitudes (from 25°N to 30°N) resulted in development of northwesterly streams over the region. Thus, unlike the wet period, less moisture is transferred into SiP during the dry period.

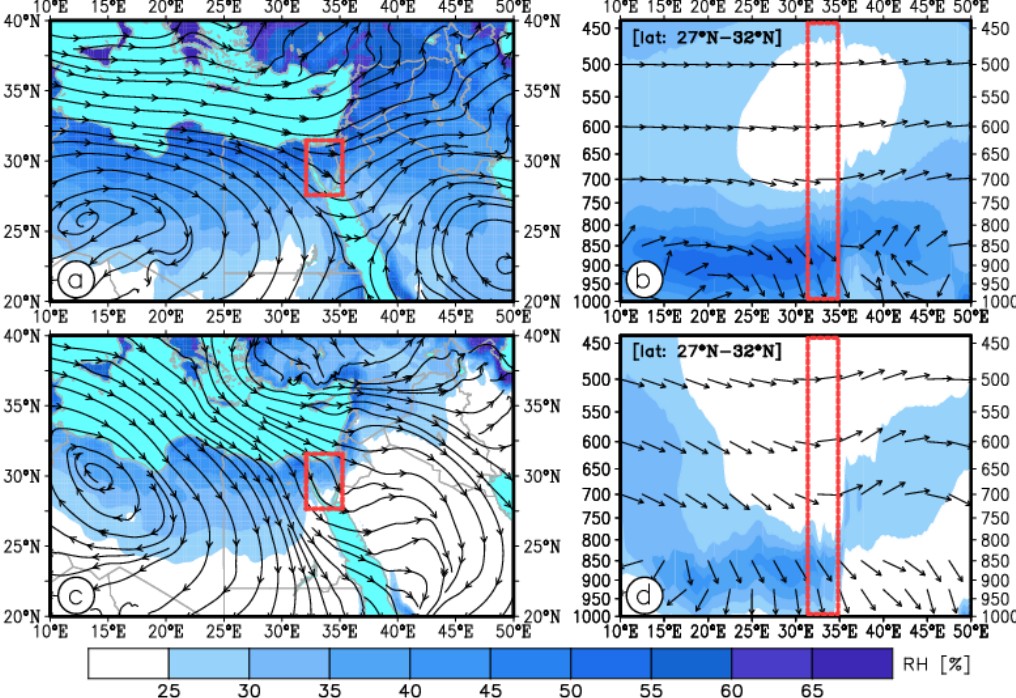

**Figure 8.** Climatology moisture condition (2001-2020) during the wet period (a, b) and dry period (c, d): panels a and c indicate 850-hPa relative humidity (RH) and wind streams; panels b and d indicate the vertical cross-sections of RH and wind profiles averaged for latitudes 27°N-32°N. Red box in the panels indicates the location of the Sinai Peninsula (SiP).

### 3.2.3 Spatial correlation analysis

In this section, daily-scale relationships of SiP's precipitation associated with the regional atmospheric variations responsible for the occurrence of wet and dry periods are explored. Figures 9a and 9b show the spatial correlation patterns between SiP's rainfall and regional see level pressure (SLP) during the wet-period and the dry-period, respectively. A negative correlation (r = -0.1) is seen over SiP. This indicates that a high association is realized between higher rainfall events (magnitude and frequency) and lower surface pressure fields over the eastern Mediterranean including SiP in the wet period (Fig. 9a). Contrariwise, a positive correlation (r = 0.25) is found between the rainfall and SLP over SiP (Fig. 9b), highlighting the dominance of high-pressure fields over the region that restrict rising of the air during the dry period. The spatial patterns at midlevel of 700-hPa also represent a negative correlation (r = -0.24) between SiP's rainfall and geopotential height (HGT) during the wet period (Fig. 9c).

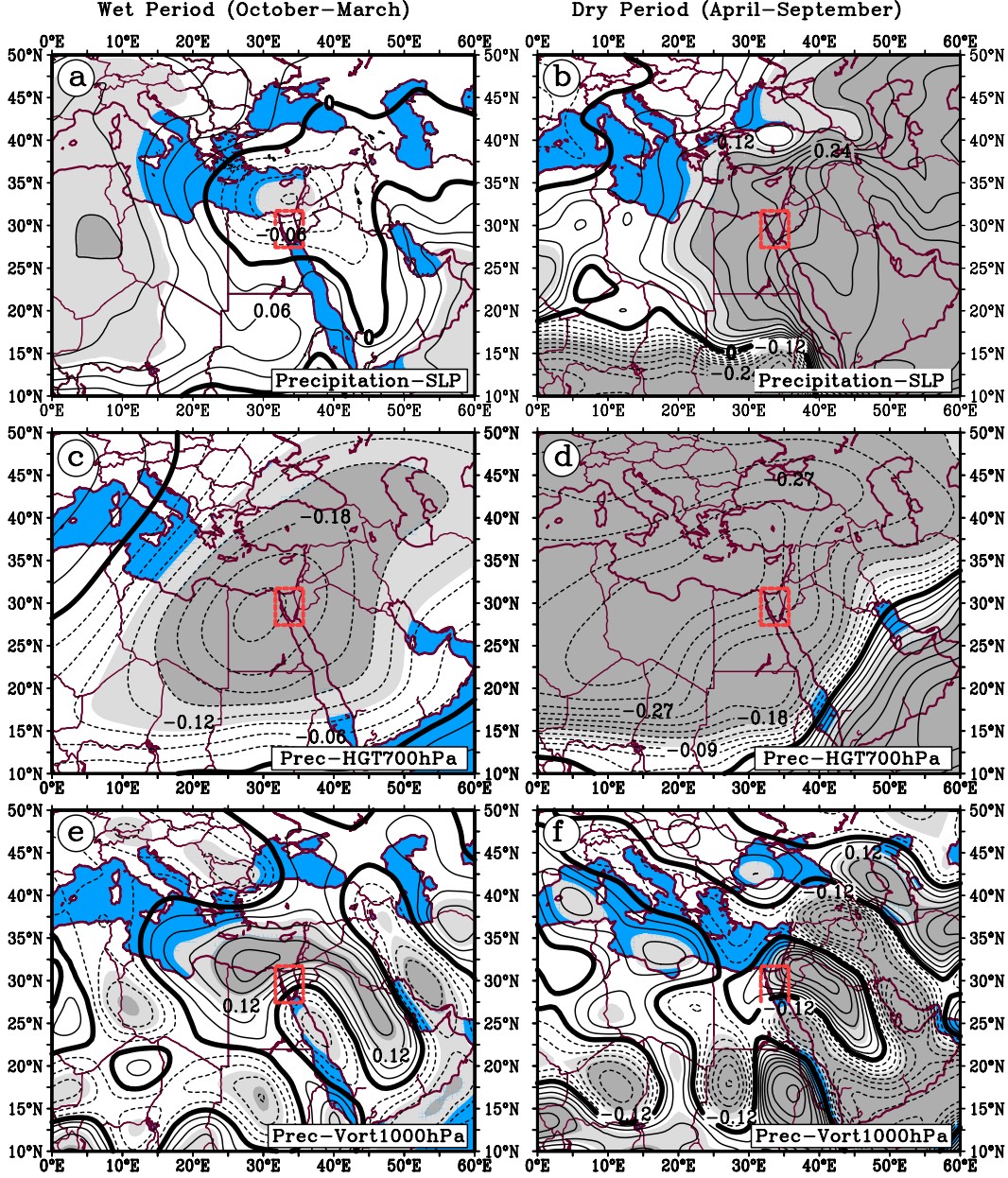

**Figure 9.** Spatial correlation patterns between the daily precipitation amount averaged over the Sinai Peninsula (SiP) (red box in each panels) and key regional atmospheric variables in the wet-period (left panels) and dry-period (right panels) for the period of 2001-2020. In each panel, the correlation is conducted between precipitation and: (a and b) SLP; (c and d) geopotential height HGT at 700-hPa; and (e and f) relative vorticity RV at 1000-hPa. The statistical significance at 95% and 99% levels are shown in light-gray and dark-gray colors, respectively.

A similar spatial pattern with a higher correlation coefficient (r = -0.3) is observed in the dry period also. However, a significant decrease in the region's rainfall could be justified by the predominance of subtropical high-pressure centers and increase of HGT during the dry period; thus, a meaningful relationship is formed between the two (Fig. 9d). The potential vorticity (PV) at the low-level of 1000-hPa correlates positively with the rainfalls in both wet and dry periods, indicating a cyclonic circulation in lower atmosphere over SiP region. However, positive PV (r = 0.12) has been dominated over the eastern Mediterranean including SiP during wet period (Fig. 9e); whereas its impact remarkably diminished over the region in the dry period (Fig. 9f), resulting in a decrease of precipitation in the eastern Mediterranean basin.

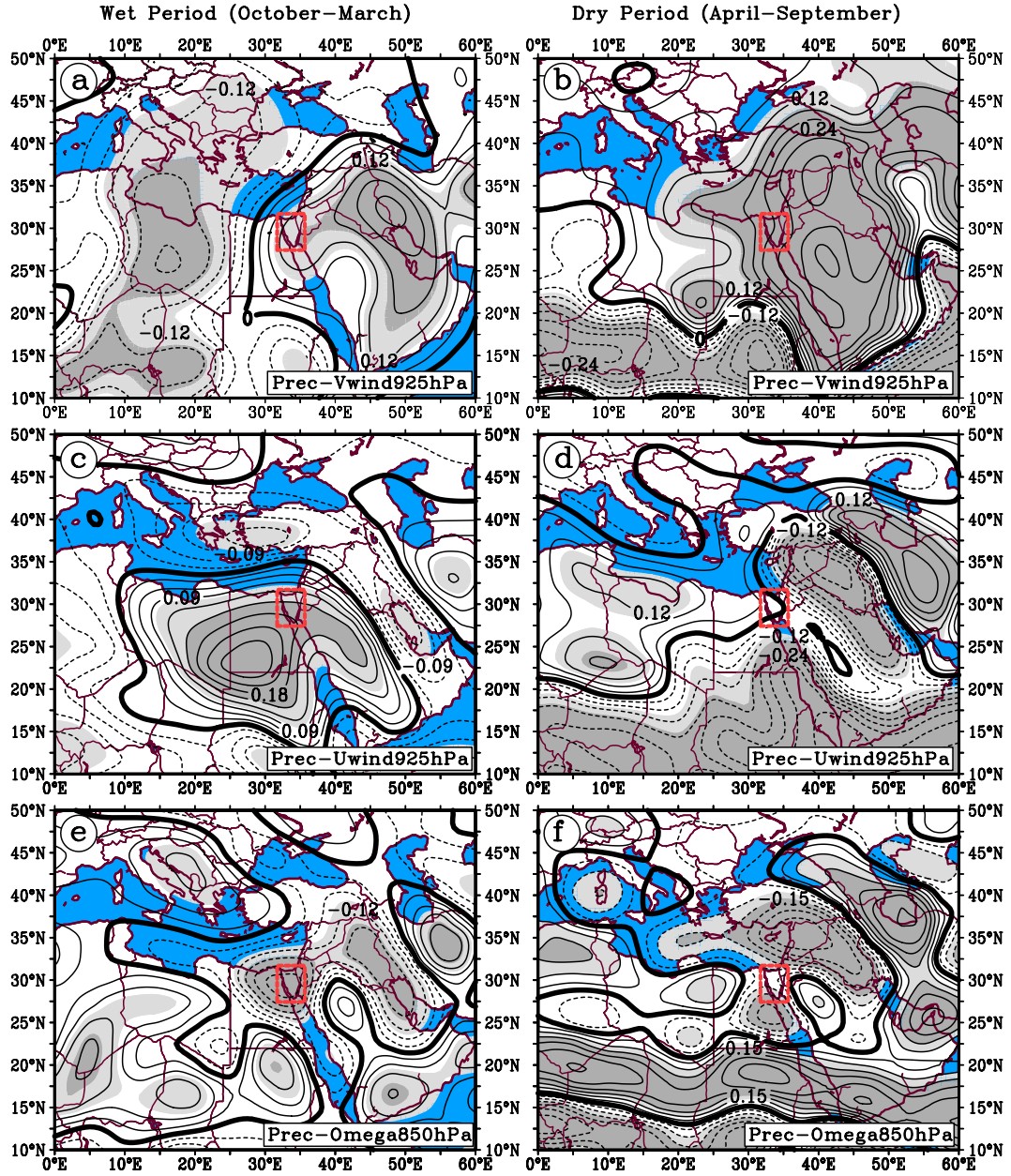

**Figure 10.** Same as Fig. 9, but for the correlations between precipitation and: (a and b) meridional wind (V-wind) at 925-hPa; (c and d) zonal wind (U-wind) at 925-hPa; and (e and f) vertical velocity (omega) at 850-hPa.

A coupling-correlation-pattern, as shown in Fig. 10, is observed with respect to the precipitation and the meridional wind (V-wind) at 925-hPa level over SiP during wet period (Fig. 10a). This indicates that, SiP's precipitation positively correlated (r = 0.12) with the southerlies found across the Middle-East with a core on the Mesopotamia, see Fig. 12 for the locations), but negatively correlated (r = -0.15) with the northerlies found over central/eastern Mediterranean and north of Africa. This provides a suitable condition for moisture transport from the Red Sea (by the southerlies) and the Mediterranean Sea (by the northerlies) into the study area. In contrast, the region is dominated by the southerly winds during dry period (Fig. 10b), which limits the role of Mediterranean to feed the region with abundant moisture, thus rain events are largely reduced. Interestingly, likewise the V-wind, a similar coupling-pattern is also observed between precipitation and zonal wind (U-wind) at 925-hPa level over the area during wet period (Fig. 10c). In such circumstances, SiP's rainfall positively correlates (r = 0.15) with westerlies over the eastern Mediterranean basin. However, in dry period (Fig. 10d), SiP's precipitation is largely associated with the negative predominant westerlies over the Mesopotamia and north of Saudi Arabia. Finally, SiP's wet period precipitation correlates negatively (r = -0.18) with the omega at lower atmosphere (at 850-hPa, Fig. 10e)

over the eastern Mediterranean basin indicating a strong vertical velocity. The relationships of SiP's rainfall and
vertical velocity are largely weakened (r = -0.08) during dry period (Fig. 10f), thus limits the rising of air to a large
extent.

## 3.3 Cyclone tracking in the wet and dry periods

Figure 11 displays the daily tracks of cyclones precipitated ≥10mm/day over SiP in wet and dry periods for the
climatology period of 2001-2020. Total numbers of cyclones during the wet and dry periods were found to be 125
and 31 cases, respectively. The cyclones of each period were classified into five categories (see Table 2) based on
the total rainfall received across SiP. During the wet period, large majority of the cyclone systems (75%) occur
within the categories of 1 and 2 (rainfall ranged 10-30mm/day). This implies that less significant storms have
struck SiP during the wet period. Yet, about 15% of the cyclones (with a rainfall >40mm/day) are potentially able
to produce torrential rainfalls, which may lead to flash floods over the region. Concerning the cyclogenesis,
Mediterranean Sea plays a significant role on either cyclogenesis -or strengthening the cyclones passing through
the area (Alpert and Shay, 1994; Flocas *et al.,* 2010; Almazroui *et al.,* 2014); this point becomes clear by looking
at Fig. 11a. However, considerable numbers of the cyclonic systems are also generated either in the North Atlantic
Ocean (then, transferred into the region via passenger cyclones) -or as the result of the Red Sea Trough (Krichak
*et al.,* 1997; de Vries *et al.,* 2013; Hochman *et al.,* 2020).

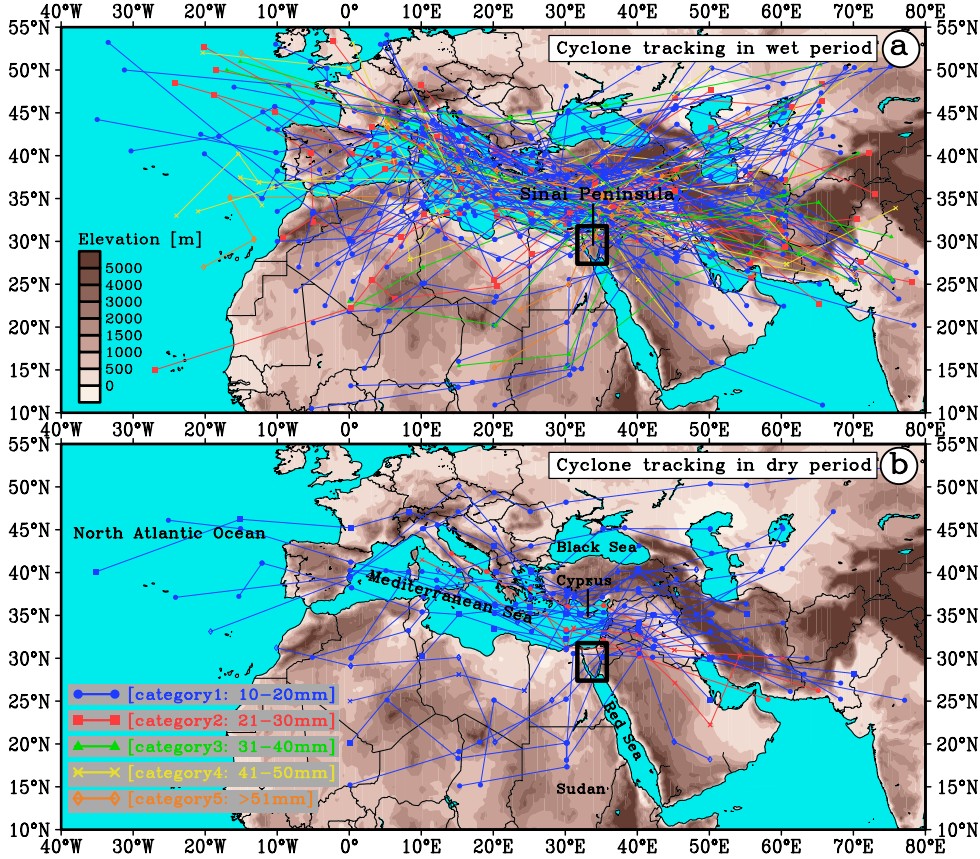

**Figure 11.** Daily track of cyclones that precipitated (≥10mm/day) over the Sinai Peninsula (SiP) during: (a) wet-period from
October to March, and (b) dry-period from April to September for the period of 2001-2020 (7305 days). Details of all cyclones
(156) classified into five categories are given in Table 2.

Figure 11b also shows the daily tracks of 31 cyclones passed through SiP region during the dry period. Unlike to
the wet period (Fig. 11a), not only the number of cyclones reduced significantly, but also their magnitudes. The
highest frequency of cyclones, according to Table 2, occurs in category 1 with 27 cyclones (87%); and followed
by only 4 cyclones (13%) in the category of 2, which have been formed within the Mediterranean (unlike category
2 of the wet period) and then moved eastwards. Interestingly, no cyclonic systems with rainfall >10mm/day taken
place within the past twenty-years during the dry period over SiP.
**Table 2.** Cyclone tracking characteristics over the Sinai Peninsula (SiP) for the period of 2001-2020.

| Cyclone classification | Total precipitation range | Frequency and percentage of cyclones | |
|---|---|---|---|
| | | Wet period | Dry period |
| Category 1 | 10-20 mm | 77 (61.2%) | 27 (87%) |
| Category 2 | 21-30 mm | 17 (13.8%) | 4 (13%) |
| Category 3 | 31-40 mm | 12 (9.7%) | - |
| Category 4 | 41- 50 mm | 10 (8.1%) | - |
| Category 5 | > 51 mm | 9 (7.2%) | - |
| - | - | 125 (100%) | 31 (100%) |

## 4 Discussion

The main focus of this study remains on quantifying the extreme precipitation events from a statistical and synoptic
perspective over SiP in the eastern Mediterranean basin over the past two-decades. SiP's literature is poor; meaning
that, although several (relavent) studies have conducted over the eastern Mediterranean (e.g. Krichak *et al.,* 1997;
Alpert *et al.,* 2002; Gabella *et al.,* 2006; Nastos *et al.,* 2013; Mathbout *et al.,* 2018; Rinat *et al.,* 2021); minimal
studies however are available over SiP, yet mostly focused on heavy rainfall-related flash floods (El Afandi *et al.,*
2013; Dadamouny and Schnittler, 2016; Arnous and Omar, 2018; Baldi *et al.,* 2020; El-Fakharany and Mansour,
2021). Thus, the novelty of this research is a combination of the satellite-reanalysis approach for a climatology
data analysis. This enabled us to quantify the precipitation characteristics (e.g. spatial patterns, spatiotemporal
variability, frequency, standard deviation, and monthly regime) and to discover the major synoptic systems (e.g.
cyclogenesis, atmospheric circulation pattern, moisture condition, spatial correlation, and cyclone tracking)
attributed to the occurrence of heavy rainfalls across SiP region.
Our statistical analysis, as one of the first analyses over SiP, revealed that distributions of the rainfall events highly
vary in time and space across SiP. From a spatial perspective, we found that the precipitation climatologies are
quite unevenly distributed across SiP. So that, norht/northeastern parts receive the highest rainfall with >100
mm/year and south/southwest parts the lowest with <30 mm/year (Fig. 2a). Using a multi-statistical-appraoch
developd in this research (Fig. 3), SiP's wet-period (October-March) and dry-period (April-September) were
determined. The outcomes from the three statistics of 90[th] percentiles, frequencies with a threshold of ≥10mm/day,
and standard deviations were in good agreement with respect to SiP's rainfall variability in time and space. Overall,
a profound dissimilarity was found in monthly precipitation values during the wet and dry periods (ranging from
5-70mm/month to 1-3mm/month, respectively); yet, thier spatial patterns were largely resembled. This means that
the rainfall amount is notably increased from south towards northeast of SiP in both periods (Fig. S10).
The EOF-based spatiotemporal variability analysis showed that the precipitation rate is insignificantly increasing
in the southern SiP (Fig. 4). This positive trend, however, may contribute to increase the occurrence of flash-flood
in the southern SiP, where a higher evelation gradient is found (see Fig. 1). Opposed to south, however, EOF
patterns (especially for cold period) revealed a severe below-average-rainfall condition (drier trend) in the north-
half of SiP; this was also captured by the anomaly-based wintertime rainfall trend (Fig. S8a). EOF analysis and
anomaly-based results are consistent with the previous findings achieved over the eastern Mediterranean basin
such as in Israel and Gaza-Strip, as pointed out by Yosef *et al.* (2009), Ziv *et al.* (2013) and Ajjur and Riffi (2020).
With respect to the temporal precipitation regime (Fig. 5), it was found that the highest monthly precipitation ratios
occur in early winter; yet mostly limited to northern SiP. This denotes that the remaining months could experience
a mild-to-severe prolonged dry-weaether (drought) condition.
Our synoptic analysis (Fig. 6) was conducted to explore the association of the synoptic systems to the precipitation
occurrence over SiP during wet and dry periods (2001-2020). Basically, majority of the cyclones (rainy systems)
affecting the study area are generated within the Mediterranean basin itself and the nearby regions, which
spatiotemporally are smaller and have shorter lifetimes compared to those of the north Atlantic systems; a similar
result was also reported by Trigo *et al.* (1999) and Buzzi *et al.* (2005). Yet, they are capable of inducing extreme
precipitation events and floods in some cases (Homar *et al.,* 2007). Accordingly, we also found that during the wet
period (Fig. 12a), two major sources of cyclonic activities (cyclogenesis) are responsible for majority of the rainfall
events over the region; these are located in the westen part (at the lee of Alps Mountains over Gulf of Genoa) and
eastern part (at the lee of Taurus Mountains over Cyprus) of the Mediterranean Sea. The cyclones formed over the
Cyprus alegedly play a significant role in the occurrence of rainfalls over the eastern Mediterranean (Saaroni *et*
*al.,* 2010). Besides, another key synoptic system that plays a secondary role in the eastern Mediterranean's rainfall
during the wet period is the Red Sea Trough, which is developed as a result of the coexistance of the Sudan's Low
and Saudi Arabia's Anticyclone (Fig. 12a). However, allegedly the Red Sea Trough has a limited contribution to
SiP's rainfall comapred to the northeastern parts of the Mediterranean basin such as over Israel (Saaroni *et al.,*
1998, and Tsvieli and Zangvil, 2005). However, during dry period (Fig. 12b), number of the Mediterranean's
cyclones are significantly reduced due to the predominance of the high-pressure systems extending from Atlantic
and north of Africa. This situation largely prevents rising of the air and, in turn, condensation, which all limit
precipitation genesis over the region during the dry period. However, as the result of the northwestwards extension
of the Persian Trough into the eastern Mediterranean basin, limited number of cyclones could develop and produce
rainfall over the eastern Mediterranean (Alpert *et al.,* 1990; Saaroni and Ziv 2000) including SiP region, as shown
in Fig. 12b.

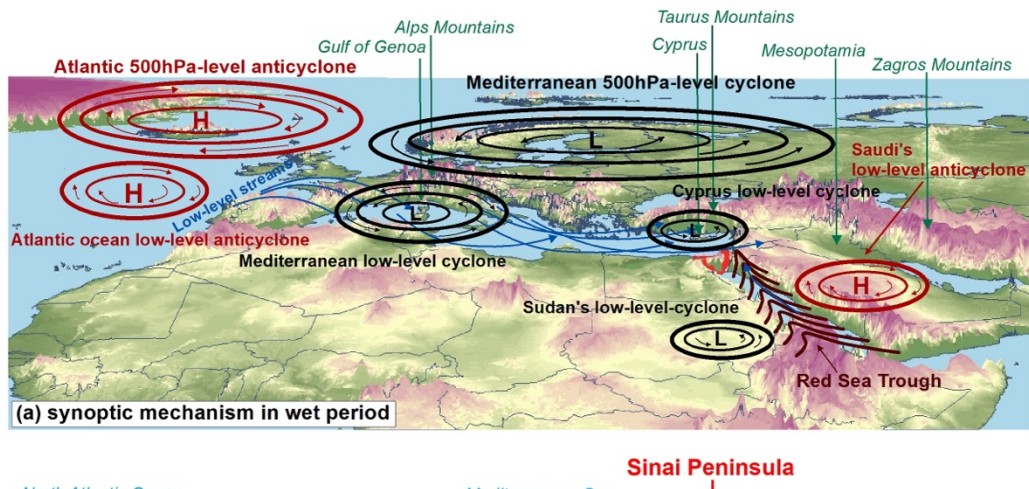

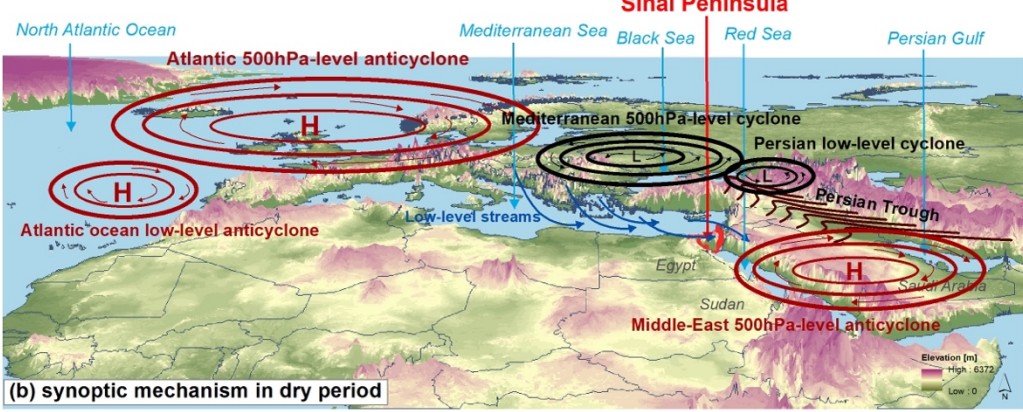

**Figure 12.** Schematic representation of the dominant synoptic systems corresponding for the precipitation events over the Sinai
Peninsula (SiP) (and the eastern Mediterranean basin) in: (a) wet-period from October to March, and (b) dry-period from April
to September for the climatology period of 2001-2020. In the maps, L and H denote the Low-pressure (cyclone) and the High-
pressure (anticyclone) systems, respectively.
With respect to the relationships of SiP's rainfall against key regional atmospheric variables (Figs. 9 and 10), we
found meaningful correlations amongst, but varied remarkably during the wet and dry periods. In this context, a
special coupling-correlation pattern was observed between SiP's rainfall against U-wind and V-wind components
in wet period. However, despite a clear association between rainfall and atmospheric variables, their correlation
coefficients were found to be relatively low (< ±0.3). A couple of major controlling factors, among others, could
explain these low r-values. First, a long timeseries of the variables in each episode (>3600 days), and second, a
very low rate of annual rainfall over SiP (on average 10-100 mm/year). Regarding the former, for instance, we did
examine with fewer timeseries (e.g., 100 days), then r-values doubled (-or tripled in some cases). Therefore,
seemingly with a longer timeseries, more smoothed correlation coefficients could be expected. It is also noted that
we found that the magnitude of correlations in the dry period are notably high. This could be explained by a semi-
stationary structure of the pressure systems over the region, which despite a low number of rainy-days, play a
crucial role in increase of r-values of the dry period compared to those of in the wet period. This implies that,
allegedly presence of the low-pressure patterns at lower atmospheric levels over the eastern Mediterranean during
the dry period of the year are well associated with lower precipitation.
Finally, a daily track of cyclones precipitated (≥10mm/day) over SiP was drawn separately for the wet period (125
cyclones, Fig. 11a) and the dry period (31 cyclones, Fig. 11b). All cyclones were classified into five categories
(see Table 2) based on the total precipitation received SiP-wide. Basically, occurrence and frequency of rainfall
events in the eastern Mediterranean region (including SiP) are largely associated with the passage of cyclonic
systems (Ulbrich et al., 2012), of which most of the cyclones are generated within the Mediterranean Sea basin in
particular during winter season (Campins et al., 2000; Nissen et al., 2010). Amongst, some cyclones are capable
of inducing extreme precipitation and floods in the region (Buzzi et al., 2005; Homar et al., 2007). We found that
about 15% of the cyclones (rainfall >40mm/day) in wet period are potentially able to produce torrential rainfalls
leading to flash floods over SiP. Unlike wet period (Fig. 11a), both number of cyclones (from 125 to 31) and their
magnitudes (from 5 to 2 categories) reduced significantly in dry period (Fig. 11b). Considering the monthly
frequency of cyclones passing through the region, during the wet period February receives the highest numbers of
cyclones with 26 out of 125 (20.8%), and followed by January (No. 25, 20%), December (No. 24, 19.2%), March
(No. 21, 16.8%), November (No. 16, 12.8%), and finally October with the lowest number of 13 (10.4%). We also
found that the cyclone of March 12, 2020 was the most significant rainy-system (>70mm/day) ever occurred in
SiP region (and perhaps in the surrounding areas) over the past two decades. This is followed by the second
extremist one occurred on December 27, 2006 with more than 62mm/day rainfall over SiP. The monthly frequency
of cyclones during the dry period also showed that April was by far the first with a total number of 20 out of 31
(65%), followed by May (No. 9, 29%), September (No. 2, 6%), and with a zero number for the rest of months (i.e.
June, July and August). Amongst, the cyclones of April 5, 2006 (27mm/day) and September 30, 2012 (24mm/day)
were found to be the extreme ones, respectively.

## 627  5 Summary

The GPM satellite remote-sensing precipitation and reanalysis NCEP/NCAR and ERA5 datasets accompanied by
a set of CDO functions and indices were employed in this research to explore extreme precipitation characteristics
over the Sinai Peninsula (SiP) particularly during wet and dry periods for the period 2001-2020. This was achieved
by i) quantifying the spatiotemporal variability, anomaly, monthly regime, frequency, standard deviation and
spatial patterns of the extreme precipitation events, ii) investigating the synoptic-scale systems responsible for the
occurrence of rainfalls, and iii) determining the major tracks of cyclones during the wet and dry periods. The key
findings are therefore summarized into three major pillars:
i.  *Spatiotemporal characteristics of rainfall:* using a multi-statistical approach based on the 90th percentiles,
frequency of days with rainfall ≥10mm/day, and spatial standard deviation SD, SiP's wet (Oct-Mar) and
dry (Apr-Sep) periods were determined. Climatology of SiP's precipitation showed that northeast and
southwest regions receive highest (>100mm/year) and lowest (<30mm/year) annual rainfall, respectively.
Also, the distribution of extreme precipitation frequencies resembled, regardless of their thresholds. This
means that highest and lowest frequencies occur in wet and dry periods, respectively. Also, trends and
patterns of the precipitation events did not show a spatiotemporal coherency across the study area, and
EOF analysis indicated a substantial drier condition in most parts especially in the northern SiP. Further,
the rainfall regime revealed that high ratios of annual precipitation and their SD are mostly estimated in
winter months.
ii. *Synoptic atmospheric systems:* the majority of cyclones precipitating over SiP are generated within the
Mediterranean basin (at leeward of the Alps and Taurus Mountains over Gulf of Genoa and Cyprus,
respectively), accompanied by the Red Sea Trough at lower levels during the wet period. These systems
either are absent or weakened significantly during dry period; however, limited lows are developed as the
result of the Persian Trough extending northwestwards. A high resemblance in the seasonal rainfall spatial
patterns (regardless of magnitude) during the wet and dry periods across SiP was observed. Also, spatial
correlations of SiP's precipitation against key regional variables at multiple levels revealed meaningful

correlation patterns, yet varied largely across the year. The relationships of SiP's rainfall against SLP, U-V winds and vertical velocity, were found to be remarkable.

iii. *Cyclone tracking:* A total number of 125 and 31 cyclones (rainfall ≥10mm/day) was tracked during the wet and dry periods, respectively. Amongst, 75% of cyclones produced rainfall ranged 10-30mm/day; while about 15% generated torrential rainfall with >40mm/day, being capable of leading to flash floods in the wet period. However, both frequency (from 125 to 31 cyclones) and magnitude (from 5 to 2 classes) of the cyclones reduced during dry period, when compared to the wet period.

**Code and data availability.** The satellite GPM, NCEP/NCAR and ERA5 reanalysis datasets used in this study are publicly available at: https://gpm.nasa.gov/, https://psl.noaa.gov/data/gridded/data.ncep.reanalysis.html, and https://www.ecmwf.int/en/forecasts/dataset/ecmwf-reanalysis-v5, respectively; the *eofs* library of Python package used herein is publicity available at: https://ajdawson.github.io/eofs/latest/index.html.

**Supplement.** The supplement related to this article is available online at: (*will be added by the journal*)

**Author contributions.** MS, BH, AM, SCD, JA and PL designed the study. MS, AM, SCD and PL developed the research goals, and MS wrote the initial manuscript. MS and AM designed and produced the figures and tables. All authors contributed to the interpretation of results and improvement of the manuscript.

**Acknowledgements.** This research was financially supported by the European centre of excellence for sustainable water technology (Wetsus). The authors would like to acknowledge the Max-Planck-Institute for Meteorology for developing the CDO-tool's functions used in this study to estimate a set of climate indices. A special thanks goes to the NASA/Goddard Space Flight Center for providing the GPM-IMERG (V06B) satellite rainfall data. We also gratefully appreciate the NOAA-NCEP/NCAR and ECMWF-ERA5 reanalysis dataset used in this research.

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
