# Peer review of "A 20-year satellite-reanalysis-based climatology of extreme precipitation characteristics over the Sinai Peninsula"

_Earth System Dynamics, 2022_

## Author Comment (AC2)

Review of the manuscript entitled "A 20-year satellite-reanalysis-based climatology of extreme precipitation characteristics over the Sinai Peninsula" by Soltani et al. In this paper, the authors try to characterize the synoptic conditions of extreme precipitation events over the Sinai region. They use the satellite-derived GPM daily precipitation and meteorological fields from the NCEP-NCAR reanalysis data during the 2001 - 2021 period. I have several concerns over the methods used by the authors in this study. Therefore, I suggest the manuscript should undergo a major revision.

We appreciate your time and consideration. The respected reviewer's comments/recommendations are clarified and addressed below.

**Specific comments:**

1. The authors claim that threshold precipitation of  $\geq 10 \text{ mm/day}$  is required to define the wet and dry periods over the Sinai region. I fail to understand the logic behind choosing this arbitrary threshold. Why don't you choose a percentile-based threshold rather than an arbitrary one? The 10 mm/day threshold also suggests that 9 mm/day is considered as dry. Is it correct? I think that you need to have separate thresholds for wet and dry events.

**Response:** We experimentally (but not arbitrary) used a threshold of  $\geq 10$ mm precipitation, after testing other thresholds like  $\geq 5$ mm and  $\geq 20$ mm, to define the overall wet-period and dry-period on a monthly basis in the Sinai Desert. For this, Figure 3 was our major reference determining the wet-months and dry-months, which is estimated based on the *frequency occurrence* of the precipitation events (but not precipitation amount only) with a threshold of  $\geq 10$ mm/day for the period of 2001-2020 (7305 days).

Indeed, we are particularly interested in to determine months with the lowest frequencies of precipitation in the Sinai. This is because, the current study is an initiative to the follow-up major Sinai's research, which is to examine the regreening impacts on the local/regional hydrometeorological process (such as precipitation recycling) in the Sinai that is currently a desert. Due to the poor Sinai's literature, this study aims at, among others, determining the months/periods with minimum -or no rainfall amount/frequency throughout the year in the Sinai in order to select the driest months to address the above-mentioned goal; particularly, estimating the rate of enhanced precipitation recycling under a vegetated-surface scenario during a naturally-dry-period of the year over the Sinai's water-limited environment.

In addition to the threshold of  $\geq 10$ mm precipitation (Fig. 3b, see below) determined in this study, we now did estimate the precipitation frequencies with thresholds of  $\geq 9$ mm/day and  $\geq 11$ mm/day (Figure 001), as test cases; this figure is attached below -- *but will not be added to the manuscript/supplement*. It is clear, their results ( $\geq 9$ ,  $\geq 10$ ,  $\geq 11$  mm/day) are almost identical; so, it is not much about thresholds with  $\pm 1$ mm/day with respect to frequencies to be defined as dry -or wet at a monthly basis. However, it will be different for much higher thresholds like  $\geq 5$ mm -or  $\geq 20$ mm in the Sinai, as shown in the supplement. Therefore, a threshold of  $\geq 10$  mm/day looks logical to define the wet -and dry periods in the case of Sinai Desert.

However, following your suggestion (and also the other respected reviewer to use standard deviation to better understand the rainfall variations), in addition to the frequency-analysis with a threshold of  $\geq 10$  mm/day, we now did estimate the monthly 90th percentile and standard deviation of the precipitation as well. So, in the revised manuscript, we now developed a multi-statistical-approach using three statistical measures based on percentile, frequency and standard deviation to determine/split wet -and dry months in the Sinai region -- as the results obtained from these statistical methods are in very good agreement in time and space (see Fig. 3 below).

So, in the revised manuscript, we now added a new subsection at the beginning to explicitly explain our approach developed in this study as follows:

2.3 Data analysis approach

2.3.1 Determining the Sinai's wet and dry periods

In this research, we are particularly aimed at, among others, determining months with the lowest (-or no rainfall) and the highest amounts/frequencies of the precipitation events throughout the year in the Sinai Peninsula. This is mainly because, it is planned as the follow-up Sinai's research, to assess regreening impacts on the local/regional hydrometeorological process such as precipitation recycling in the Sinai Desert under a vegetated-surface scenario during a naturally dry period of the year. Thus, herein we developed a multi-statistical-approach to split the wet -and dry months of the year in the Sinai's water-limited environment for the period of 2001-2020. This is achieved via a combination of the results obtained from three statistical measures: i) monthly 90th percentile of the daily precipitation (Fig. 3a), ii) an experimentally-based precipitation frequency occurrence with a threshold of  $\geq$ 10 mm/day (Fig. 3b) – after examining other thresholds such as  $\geq$ 5 mm/day and  $\geq$ 20 mm/day (see Figs. S1 and S2), and iii) monthly rainfall standard deviation SD estimates (Fig. 3c). These methods were calculated using a set of statistical functions, described in the follow-up subsection (see Table 1). Therefore, using the above-mentioned approach developed in this study, the Sinai's wet months are determined from October to March, defined as wet-period, and its dry months from April to September, defined as dry period.

---

## Author Response (AR1)

This study with a title of "A 20-year satellite-reanalysis-based climatology of extreme precipitation characteristics over the Sinai Peninsula" has been seriously reviewed. The authors have comprehensively quantified the extreme precipitation characteristics over the Sinai Desert in Egypt, and explored the synoptic systems responsible for the occurrence of precipitation events along with the major tracks of cyclones during the wet and dry periods. This study is of interesting and importance for understanding extreme precipitation events in this region. However, I have several comments and suggestion for this paper before its acceptance, and I would like to give a chance for moderate revision. Please see below for the details.

We appreciate your time and consideration. The respected reviewer's comments/recommendations are clarified and addressed below.

Despite that the authors have stated that the GPM data has been evaluated and employed in the Mediterranean region, I do not know whether the GPM has a better performance in the Sinai Peninsula. If the GPM has been assessed in this region, you can cite the literature for proving the capacity of this data. If not, it is better to conduct an evaluation of the GPM performance in the Sinai Peninsula. Because it is foundational for this study about the analyses of the extreme precipitation.

**Response:** To the best of our knowledge, the GPM data has not used over the Sinai Desert to date. We appreciate the reviewer's suggestion; however, it is not feasible for us to evaluate the GPM performance in the Sinai Peninsula. This is because, there are a very-limited number of weather stations in the Sinai Desert; yet, we do not have access to those limited *in-situ* data (such as precipitation) measured at the ground stations. Therefore, for our analysis in the Sinai, we relied on the satellite remote-sensing GPM precipitation data (among other global RS-Prec dataset such as CHIRPS and TRMM), as its performance has been already acknowledged by other studies over the surrounding-areas in the eastern Mediterranean region (e.g., Retalis *et al.,* 2018; Petracca *et al.,* 2018; Caracciolo *et al.,* 2018; Cinzia Marra *et al.,* 2019; Hourngir *et al.,* 2021).

It is strange to use the observations at three sites to explore the annual and seasonal changes in precipitation trend. Could they indeed represent the whole region for southern, middle and southern parts? I cannot believe that, because the precipitation have huge differences regionally. When you finish the GPM evaluation, you can use the regional mean GPM values to study the annual and seasonal changes in precipitation trend.

**Response:** The reviewer has a point. However, the site-scale annual/seasonal trends analysis in climate data (e.g., precipitation) is not unusual, and numerous studies exist in the literature (e.g., Aguilar et al. 2005; Alexander et al. 2006; Choi et al. 2009; Dos Santos et al. 2010; Soltani et al. 2016). Accordingly, for this anomaly analysis (which, is now moved to the supplement data: Fig. S8), we selected three sites in the north, south and middle of the Sinai area as good representatives for those regions. We believe that it is a meaningful analysis, as the climatology average precipitation map (2001-2020) (Fig. 4a: now Fig. 2a in the revised manuscript) over the Sinai clearly indicates that the highest, lowest and average amounts of precipitation are received in the north, south and middle regions, respectively. Unlike the heterogeneous/mountain areas with complex climate mechanisms where the precipitation varies greatly in a short distance horizontally -or vertically (such as Alpine/pre-Alpine regions), in dry regions like the Sinai Desert, this is not much the case where the precipitation almost smoothly increases from south to north -- this is true not only in the climatology map (Fig. 2a), but also in the monthly -and daily events as shown in Figs. 2b-c. Thus, it makes sense to use site observations to explore the climatology trends of precipitation in different parts of the Sinai Desert.

However, since we now applied EOF-based analysis that considers both spatial -and temporal changes in a given variable (e.g. precipitation) for both annual and biannual (Fig. 4 in the revised manuscript, *annual-scale map shown below*), as suggested by the other respected reviewer, we decided to move the temporal site-scale anomaly-based analysis into the supplement data (Fig. S8). It is good mentioning that, the results of the site-scale anomaly-based analysis are in good agreement with those of the grid-scale EOF-based spatiotemporal analysis across the Sinai.

[Figure]

**Figure 4.** The two leading EOF spatial patterns (a and c) and the associated timeseries (b and d) of the monthly mean precipitation dataset (**at annual scale**) for the period of 2001-2020 (240 months) in the Sinai Peninsula. The values of EOFs (a and c) are expressed as correlation coefficients.

In figures 7, and 9, there are so many lines to weaken the readability of the two figures. You can remove the country lines, and remain the boundary for the study region.
**Response:** Following your suggestion, these figures (now Figs. 6, 9 and 10) are modified in the revised manuscript.

The moisture condition plays a quite important role in (extreme) precipitation events, but the authors seem to omit the analysis of it. For example, in figure 7, the climatological condition of moisture during wet-period and dry-period should be included.
**Response:** Thanks for a good suggestion. In the revised manuscript, we now added Figure 8 (also shown below) to explore the climatology (2001-2020) moisture condition and wind structure at multiple-levels of the atmosphere during the wet -and dry periods over the Sinai region -and in the nearby areas.

Line 425-246: The authors said "This provides a suitable condition for moisture transport". But the low-level wind fields do not denote the moisture transport directly.
**Response:** Basically, when the low-level streams blowing over large waterbodies such as Mediterranean Sea and Red Sea, it is assumed that a considerable amount of water-vapor is transferred towards the target regions ahead. However, to avoid the guesstimate, we also did estimate the wind streams along with the moisture content condition (see below) in the revised manuscript.

[Figure]

**Figure 8.** Climatology moisture condition (2001-2020) during wet period (a, b) and dry period (c, d): panels a and c indicate 850-hPa relative humidity (RH) and wind streams; panels b and d indicate the vertical cross-sections of RH and wind profiles averaged for the latitudes 27°N-32°N. Red box in the panels indicates the location of the Sinai Peninsula.

The low-level moisture flux is supposed to be added in Fig 9.
It is believed that estimating the correlation between the rainfall and moisture (-or relative humidity) is not necessary (-or even does not make much sense). Because in order to have condensation process and precipitation event, the maximum moisture content (~100%) must be available at the lower atmospheric PBL. However, the other way around may not be necessarily true meaning that despite a considerable moisture content to be present in the atmospheric layers, it may not rain – which, could be due to the synoptic-dynamical condition (moisture availability in the Sinai's dry-period is a good example – see figure above). So, indeed the relationship between precipitation occurrence and atmospheric moisture content availability is already obvious.

In section 3.3, the authors only discussed the frequencies of different cyclones that posed various amount precipitation on Sinai Peninsula, but ignoring the tracks and intensities of different cyclones. The detailed characteristics of cyclones that affecting Sinai Peninsula are recommended to be shown in 3.3. Moreover, the characteristics of cyclones under the synoptic patterns and atmospheric circulations during wet or dry period should be analyzed in detail. So, I strongly suggest that the authors can try to analyze the characteristics of the cyclones with negative/positive precipitation. Additionally, in section 3.2, are there links between the cyclones and the anomalous circulation background?
**Response:** In Sect. 3.3, the major cyclone-tracking results are explained briefly, as it was aimed to keep it concise. However, to avoid repetition there, additional details and characteristics associated with the different cyclones are further discussed in Sect. 4 (Discussion).

Thank you for the suggestion. However, these are out of the scope of this study, -and would be a very lengthy-paper to consider and provide all the details on the types of cyclones under different atmospheric and anomalous conditions. We believe that, these additional investigations/analyses could be conducted in the separate studies. Nevertheless, we have already provided further details of the cyclones (monthly frequencies, intensity classifications, etc.,) in the Discussion section of the manuscript together with the

characteristics related to cyclone-tracking given in Table 2. What is more, there are some related studies focused only on these topics in the literature of the eastern Mediterranean basin (e.g., Alpert and Shay, 1994; Flocas *et al.,* 2010; Almazroui *et al.,* 2014; Nastos et al. 2018; Lionello et al. 2019; Hochman *et al.,* 2020), as stated in the manuscript.

Line 450-451: "This implies that less significant storms have struck the Sinai during wet period." What does the "less significant storms" mean?

**Response:** Since, 75% of the cyclones precipitated over the Sinai fall in 1 and 2 categories (out of 5) indicating insignificant rainy-systems, according to table 2; thus, the remaining categories of 3, 4 and 5 (25% all together) are considered as strong -and significant storms due to a higher potential for making flashfloods over the region damaging the society and environment. However, as stated, mostly (75%) insignificant / less significant storms have struck the Sinai during the wet season for a 20-year period from 2001 to 2020.

Review of the manuscript entitled "A 20-year satellite-reanalysis-based climatology of extreme precipitation characteristics over the Sinai Peninsula" by Soltani et al. In this paper, the authors try to characterize the synoptic conditions of extreme precipitation events over the Sinai region. They use the satellite-derived GPM daily precipitation and meteorological fields from the NCEP-NCAR reanalysis data during the 2001 - 2021 period. I have several concerns over the methods used by the authors in this study. Therefore, I suggest the manuscript should undergo a major revision.

We appreciate your time and consideration. The respected reviewer's comments/recommendations are clarified and addressed below.

Specific comments:
1. The authors claim that threshold precipitation of >= 10 mm/day is required to define the wet and dry periods over the Sinai region. I fail to understand the logic behind choosing this arbitrary threshold. Why don't you choose a percentile-based threshold rather than an arbitrary one? The 10 mm/day threshold also suggests that 9 mm/day is considered as dry. Is it correct? I think that you need to have separate thresholds for wet and dry events.

**Response:** We experimentally (but not arbitrary) used a threshold of ≥10mm precipitation, after testing other thresholds like ≥5mm and ≥20mm, to define the overall wet-period and dry-period on a monthly basis in the Sinai Desert. For this, Figure 3 was our major reference determining the wet-months and dry-months, which is estimated based on the *frequency occurrence* of the precipitation events (but not precipitation amount only) with a threshold of ≥10mm/day for the period of 2001-2020 (7305 days).

Indeed, we are particularly interested in to determine months with the lowest frequencies of precipitation in the Sinai. This is because, the current study is an initiative to the follow-up major Sinai's research, which is to examine the regreening impacts on the local/regional hydrometeorological process (such as precipitation recycling) in the Sinai that is currently a desert. Due to the poor Sinai's literature, this study aims at, among others, determining the months/periods with minimum -or no rainfall amount/frequency throughout the year in the Sinai in order to select the driest months to address the above-mentioned goal; particularly, estimating the rate of enhanced precipitation recycling under a vegetated-surface scenario during a naturally-dry-period of the year over the Sinai's water-limited environment.

In addition to the threshold of ≥10mm precipitation (Fig. 3b, see below) determined in this study, we now did estimate the precipitation frequencies with thresholds of ≥9mm/day and ≥11mm/day (Figure 001), as test cases; this figure is attached below -- *but will not be added to the manuscript/supplement.* It is clear, their results (≥9, ≥10, ≥11 mm/day) are almost identical; so, it is not much about thresholds with ±1mm/day with respect to frequencies to be defined as dry -or wet at a monthly basis. However, it will be different for much higher thresholds like ≥5mm -or ≥20mm in Sinai, as shown in the supplement. Therefore, a threshold of ≥10 mm/day looks logical to define wet -and dry periods in the case of Sinai Desert.

However, following your suggestion (and also the other respected reviewer to use standard deviation to better understand the rainfall variations), in addition to the frequency-analysis with a threshold of ≥10 mm/day, we now did estimate the monthly 90th percentile and standard deviation of the precipitation as well. So, in the revised manuscript, we now developed a multi-statistical-approach using three statistical measures based on percentile, frequency and standard deviation to determine/split wet and dry months in Sinai region -- as the results obtained from these statistics are in very good agreement temporally and spatially (see Fig. 3 below).

So, in the revised manuscript, we now added a new subsection at the beginning to explicitly explain our approach developed in this study as follows:

======================
**2.3 Data analysis approach**
2.3.1 Determining the Sinai's wet and dry periods

In this research, we are particularly aimed at, among others, determining months with the lowest (-or no rainfall) and the highest amounts/frequencies of the precipitation events throughout the year in the Sinai Peninsula. This is mainly because, it is planned as the follow-up Sinai's research, to assess regreening impacts on the local/regional hydrometeorological process such as precipitation recycling in the Sinai Desert under a vegetated-surface scenario during a naturally dry period of the year. Thus, herein we developed a multi-statistical-approach to split the wet -and dry months of the year in the Sinai's water-limited environment for the period of 2001-2020. This is achieved via a combination of the results obtained from three statistical measures: i) monthly 90th percentile of the daily precipitation (Fig. 3a), ii) an experimentally-based precipitation frequency occurrence with a threshold of ≥10 mm/day (Fig. 3b) – after examining other thresholds such as ≥5 mm/day and ≥20 mm/day (see Figs. S1 and S2), and iii) monthly rainfall standard deviation SD estimates (Fig. 3c). These methods were calculated using a set of statistical functions, described in the follow-up subsection (see Table 1). Therefore, using the above-mentioned approach developed in this study, the Sinai's wet months are determined from October to March, defined as wet-period, and its dry months from April to September, defined as dry period.

==========================================

[Figure]

**Figure 001.** Frequency occurrence of the monthly precipitation events for the climatology period of 2001-2020 (7305 days) over the Sinai Peninsula: **left)** with a threshold of *≥9 mm/day*, **right)** with a threshold of *≥11 mm/day*. Units are frequency in days.

[Figure]

**Figure 3.** A multi-statistical analysis of the precipitation in a monthly basis: a) the 90[th] percentile of rainfall climatology, b) frequency occurrence of the rainfall events with a threshold of ≥10 mm/day, and c) grid-based standard deviation SD estimates of the rainfall for the period of 2001-2020 (240 months) over the Sinai Peninsula.

2. Cyclone Tracking: From the description, the cyclone tracking method is not clear. How did they identify the genesis and lysis of cyclones? If multiple cyclones are present, how did they identify each of them at the subsequent time steps? Did the authors use an automated algorithm? In that case, it should be mentioned explicitly. There are several cyclone tracking algorithms available. The authors can compare their technique with some of the other tracking techniques.

**Response:** The approach we used for the cyclone tracking was not an automated algorithm, but a manual approach developed in this study. Each method, however, has its own pros and cons. For example, the available algorithms (in case of being freely accessible for the public) might be convenient for users to use, but need several inputs and are still suffering from a low-accuracy/precision in identifying/tracking cyclones (and yet, mostly focused on the tropical cyclones). This could be a critical challenge to be used in heterogeneous landscapes with a complex atmospheric PBL in mid-latitudes like the Mediterranean region. As such, the performance of those algorithms remains challenging for several reasons mentioned above -and also outlined in e.g., Raible et al., 2007; Flaounas et al., 2014; Prantl et al., 2021. However, our developed manual method might be a bit time-consuming, but possesses a higher degree of precision in identifying/tracking the cyclones (and also anticyclones) – as explained below.

Also, we would like to mention that the manual approach developed in this study outweighs the available automated algorithms for the Sinai's case; apart from the above-mentioned issues for using automated algorithms, we were merely interested to identify/track those rainy-systems (cyclones) that precipitated ≥10mm over Sinai – some cyclones/lows may not necessarily make rainfall at all especially with a given threshold over a given domain. And yet, as you mentioned correctly, multiple cyclones might be presented simultaneously that is an additional challenge. So, in our case this is even further challenging to use such algorithms for cyclone tracking over the Sinai, as a small-region from a synoptic/dynamical perspective.

In the revised manuscript, we now added further details to explicitly explain our approach developed for the cyclone tracking analysis in this study as follows:

==================

2.3.5 Cyclone tracking
In line with the synoptic analysis, the daily trajectories of the rainy-systems precipitated over the Sinai region were tracked and plotted for the wet -and dry periods using a manual approach developed in this study. In our approach, we merely aimed to detect and track cyclones precipitated with ≥10mm in the Sinai. This however is challenging for an automated algorithm to a large extent, to detect a low-pressure-system (sometimes with multiple centers, cyclones) that may -or not has generated a rainfall with a given threshold over a given domain. Yet, its performance is not error-free in particular over heterogeneous regions with complex atmospheric PBL like the Mediterranean region (e.g., Raible *et al.,* 2007; Flaounas *et al.,* 2014c; Prantl *et al.,* 2021). Our manual-based cyclone-tracking approach developed in this study consists of three major steps as follows:
*i) first*, a set of daily total precipitation patterns over the Sinai was produced using GPM data separately for the wet -and dry periods; by doing so, a total number of 156 events (out of 7305 days) were identified, for which precipitated ≥10mm over the Sinai Peninsula. Accordingly, synoptic-scale daily composites of SLP, and 850-hPa RV and streamflow were produced using the reanalysis data for the entire study-period (2001-2020, 7305 days). Here, 850-hPa RV and streamflow were used along with SLP to better identify the lows (Flaounas *et al.,* 2014c). *ii) second*, to identify the cyclogenesis/lysis of the selected events, the composite maps of SLP, RV and streamflow for several days before -and after the Sinai's rainfall events were monitored and tracked with care. Every daily movement (X-Y coordinates) of the corresponding cyclone was recorded from the beginning (where the low system was born, cyclogenesis) until it was disappeared (cyclolysis). This process was carried out one-by-one for all156 cases with rainfall ≥10mm. All the events were classified into five categories based on the rainfall magnitude as follows: category 1 (10-20mm), category 2 (21-30mm), category 3 (31-40mm), category 4 (41-50mm) and category 5 (>51mm). *iii) third*, finally the cyclone tracking charts for the wet -and dry periods were produced using the information obtained from the former steps.
==========================================================

3. Statistical significance of the trends: The authors should do a significance test (ideally, a non-parametric test) for the trends presented in Fig. 2 and report it in the caption.

**Response:** The statistical tests for significance of the regression model/trends are typically performed for the time-series dataset (between two variables), which can be determined by e.g. $r^2$-values along with the p-values (or t-test). As such, if $r^2$ is typically >0.6 with p <0.05 in regressions, then the trend is regarded as statistically significance and meaningful. However, our trend of slope (Fig.2: now Fig. S8 in the revised supplement) is estimated using an *anomaly-based approach* (not timeseries in a regression model). Here, we detected the precipitation anomalies (annual/seasonal) with a *Mean* function, meaning that the long-term mean (average) of each rainfall data was calculated; then it was subtracted from each year/season precipitation values to estimate the anomalies (i.e. anomaly equals individual values of each year/season minus long-term mean value), and finally the trend of slopes of those anomalies was drawn using the common least-squares fitting process. Therefore, our trend of slope represents the rate at which change occurs over time. If the slope has a positive value, the rainfall rate is increasing (-or the wetter condition). If it is negative, the rainfall rate is decreasing (-or the drier condition). In that figure, we interpreted the trend of slope to mean that, on average, the rainfall rate is changed by the slope value each year/season over the past two decades (2001-2020). Therefore, we deliberately avoided to use the terms such as ''significant trend'' -or ''statistically significance'', related to that figure.

Indeed, for the *anomaly-based approach* and its trend of slopes it is not feasible to perform any kind of statistical significances like Bootstrapping; however, we did perform *Bootstrapping Confidence Interval* for the *original datasets* of seasonal and annual precipitation climatology (20-years) on the selected sites across the Sinai. The results are given in Table S1 in the revised supplement – also attached below.

However, since we now applied the EOF-based analysis (Fig. 4 in the revised manuscript, also shown below) suggested by you, we decided to move the temporal site-scale anomaly-based analysis/figure into the supplement data (Fig. S8). It is good mentioning that, the results of the site-scale anomaly-based analysis are in good agreement with those of the grid-scale EOF-based spatiotemporal analysis in the Sinai Desert.

**Table S1.** The 95% and 99% bootstrapped confidence interval (CI) for the Mean and Median values of the original dataset (mean seasonal and annual for 20-years: 2001-2020) for the selected sites across the Sinai (anomaly-based analysis in Fig. S8), see Fig. 1 for the locations. For this analysis, 300 bootstrapped samples were generated each with a sample size of n=10.

| | North-site | | | Middle-site | | | South-site | | |
|---|---|---|---|---|---|---|---|---|---|
| | Winter | Autumn | Annual | Winter | Autumn | Annual | Winter | Autumn | Annual |
| *Average precipitation (mm)* | *68.6* | *18.5* | *28.4* | *22* | *6.1* | *9.3* | *9.1* | *4.8* | *5.1* |
| **95%** bootstrapped CI for **Mean** value of original dataset | 79.8 | 24 | 31.9 | 30.8 | 9.1 | 11.2 | 14.3 | 9.4 | 6.9 |
| **99%** bootstrapped CI for **Mean** value of original dataset | 85.5 | 26 | 33.4 | 35.4 | 10.3 | 12 | 16.2 | 10.8 | 7.3 |
| **95%** bootstrapped CI for **Median** value of original dataset | 88.3 | 26.1 | 31.4 | 23.9 | 7.5 | 10.5 | 11.7 | 5.3 | 7.5 |
| **99%** bootstrapped CI for **Median** value of original dataset | 90.4 | 29.1 | 36.1 | 29.1 | 9.7 | 12 | 17.5 | 8.2 | 8.3 |

4. Fig. 4: I don't understand the logic behind this analysis. Why do you need to compare the annual mean precipitation with the wettest month and wettest day precipitation? The color scales of all the plots should be the same for comparison. There are better ways for understanding spatio-temporal variability. E. g. an EOF analysis.

**Response:** The reason is to demonstrate how precipitation is *spatially* distributed across the Sinai, not only in the climatology map (20-years -or 240 months -or 7305 days) (Fig. 4: now Fig. 2a in the revised manuscript) but also in a single month (Fig. 2b) -or day (Fig. 2c). This study focuses on extreme events; thus, the most extreme cases of the wet month/day are presented as well, and more extreme cases of the rainiest months (Fig. S3) and wettest days (Fig. S4) are represented in the revised supplement data. All panels in Figure 2a-c clearly show that northern (southern) parts of the Sinai *always* receive the highest (lowest) amount of precipitation regardless of time-period *either* in long-term climatology *or* a single month/day-event. However, we now re-structured the associated subsection to (*3.1.1 The precipitation spatial patterns and extreme indices*) in the revised manuscript. Please also note that Figure 6 (climate extreme indices) is now attached to Figure 2d-f, due to the new subsection added – see Figure 2 below.

We needed to use two separated legends for the panels *a* and *b-c* in Figure 2 due to large differences in the rainfall magnitudes. In case of using a single legend, the distribution of rainfall values in panels *b* and especially *c* will become almost constant/flat, thus unclear.

Thanks for a good suggestion. We now applied the Empirical Orthogonal Function (EOF) analysis to better understanding the spatiotemporal variability of the Sinai's precipitation climatology. This analysis -and its results for both the annual and seasonal (wet -and dry periods) have been added to the revised manuscript (Fig. 4) and supplement data (Figs. S6 and S7) -- these figures are shown below also.

[Figure]

**Figure 2.** The precipitation spatial patterns and extreme indices: a) climatology map of mean annual precipitation (2001-2020); b) the wettest month i.e. March 2020 (out of 240 months), c) the wettest day i.e. March 12, 2020 (out of 7305 days); as well as extreme daily precipitation indices with a threshold of ≥1mm/day: d) simple daily intensity index (SDII), e) consecutive dry days (CDD) and f) wet days index (RR1) for the period of 2001-2020 over the Sinai Peninsula.

[Figure]

**Figure 4.** The two leading EOF spatial patterns (a and c) and the associated timeseries (b and d) of the monthly mean precipitation dataset (**at annual scale**) for the period of 2001-2020 (240 months) in the Sinai Peninsula. The values of EOFs (a and c) are expressed as correlation coefficients.

[Figure]

**Left: Figure S6.** Same as in Fig. 4, but for the wet-period (October-March). **Right: Figure S7.** Same as in Fig. 4, but for the dry-period (April-September).

Review of "A 20-year satellite-reanalysis-based climatology of extreme precipitation characteristics over the Sinai Peninsula" by Soltani et al. (Submitted to ESD). Summary and Recommendation: This study invoked IMERG precipitation and NCEP/NCAR reanalysis dataset for atmospheric variables to first identify extreme rainfall characteristics followed by understanding synoptic properties responsible for wet and dry periods observed over Sinai Peninsula. The authors use a range of tools in CDO toolbox to perform statistical analysis over the region. The study has merit in terms of identifying mechanisms responsible for extreme rainfall events but it needs more statistical basis so as to establish "meaningful" relationship between atmospheric state and precipitation events. I highly recommend not using strong sentences such as "remarkable correlation" and "meaningful results" without performing some kind of statistical significance tests on their results. I also found out many spelling and grammatical mistakes with incoherency in their sentences throughout the manuscript and it was impossible for me to pin point each of the error and thus I highly recommend going through the manuscript carefully to fix those errors before submitting the revised version of this manuscript. Therefore, I recommend "Major Revision" before I can recommend accepting this manuscript.

We appreciate your time and consideration. The respected reviewer's comments/recommendations are clarified and addressed below.

My primary suggestions are as follows:

1) I suggest adding the lat-lon bounds of the entire study region Sinai Peninsula corresponding to their Figure 1 Description.

**Response:** It is done in the revised manuscript.

2) Multiple spelling and grammatical errors are present throughout the manuscript and thus I cannot pin point each of them, so please correct those throughout the manuscript.

**Response:** It is done in the revised manuscript.

3) Lines 137-139: I am not sure if I agree with this statement. I have observed cyclonic and anticyclonic patterns in coarser and finer resolution with almost similar accuracy and it was even better captured in finer resolution. I do not mind authors using coarser resolution product for their analysis but this statement is not necessarily true and I suggest removing this from their manuscript.

**Response:** The reviewer has a point; however, a coarser resolution data to diagnose/explore large-scale pressure-fields (cyclones/anticyclones) is indeed wise to be used in regions with a complex atmospheric PBL such as the eastern Mediterranean basin. This is particularly true for the low-level features (e.g. sea level pressure SLP, relative vorticity RV, potential vorticity PV) due to the strong interactions between the region's orography and low-level PBL dynamics.

However, following the suggestion, we now modified the sentence as follows in the revised manuscript: *''... NCEP/NCAR data was used to study the pressure fields due to its coarser resolution, as it is believed that large-scale pressure systems such as cyclonic -and anticyclonic patterns could be better represented in a coarse resolution especially at lower levels of the atmosphere over complex regions''*.

4) Line 170: How did authors come up with the threshold of 10 mm/day for this region? Are there previous studies available backing this claim or did authors perform any statistical analysis to come up with this threshold? Currently this looks like an arbitrary threshold and I don't think I can accept this as it is.

**Response:** We experimentally (but not arbitrary) used a threshold of ≥10mm precipitation, after testing other thresholds like ≥5mm and ≥20mm, to define the overall wet-period and dry-period on a monthly basis in the Sinai Desert. For this, Figure 3 was our major reference determining the wet-months and dry-months, which is estimated based on the *frequency occurrence* of the precipitation events (but not precipitation amount only) with a threshold of ≥10mm/day for the period of 2001-2020 (7305 days).

Indeed, we are particularly interested in to determine months with the lowest frequencies of precipitation in Sinai. This is because, the current study is an initiative to the follow-up major Sinai's research, which is to examine the regreening impacts on the local/regional hydrometeorological process (such as precipitation recycling) in the Sinai that is currently a desert. Due to the poor Sinai's literature, this study aims at, among others, determining the months/periods with minimum -or no rainfall amount/frequency throughout the year in the Sinai in order to select the driest months to address the above-mentioned goal; particularly, estimating the rate of enhanced precipitation recycling under a vegetated-surface scenario during a naturally-dry-period of the year over the Sinai's water-limited environment.

In addition to the threshold of ≥10mm precipitation (Fig. 3b, see below) determined in this study, we now did estimate the precipitation frequencies with thresholds of ≥9mm/day and ≥11mm/day (Figure 001), as test cases; this figure is attached below -- *but will not be added to the manuscript/supplement*. It is clear, their results (≥9, ≥10, ≥11 mm/day) are almost identical; so, it is not much about thresholds with ±1mm/day with respect to frequencies to be defined as dry -or wet at a monthly basis. However, it will be different for much higher thresholds like ≥5mm -or ≥20mm in the Sinai, as shown in the supplement. Therefore, a threshold of ≥10 mm/day looks logical to define the wet -and dry periods in the case of Sinai Desert.

However, following the other respected reviewer to use percentile-based approach (and your suggestion to use standard deviation to better understand rainfall variations), in addition to the frequency-analysis with a threshold of ≥10 mm/day, we now did estimate the monthly $90^{th}$ percentile and standard deviation of the precipitation as well. So, in the revised manuscript, we now developed a multi-statistical-approach using three statistical measures based on percentile, frequency and standard deviation to determine/split wet and dry months in Sinai region -- as the results obtained from these statistics are in very good agreement temporally and spatially (see Fig. 3 below).

So, in the revised manuscript, we now added a new subsection at the beginning to explicitly explain our approach developed in this study as follows:

============================

**2.3 Data analysis approach**
2.3.1 Determining the Sinai's wet and dry periods
In this research, we are particularly aimed at, among others, determining months with the lowest (-or no rainfall) and the highest amounts/frequencies of the precipitation events throughout the year in the Sinai Peninsula. This is mainly because, it is planned as the follow-up Sinai's research, to assess regreening impacts on the local/regional hydrometeorological process such as precipitation recycling in the Sinai Desert under a vegetated-surface scenario during a naturally dry period of the year. Thus, herein we developed a multi-statistical-approach to split the wet -and dry months of the year in the Sinai's water-limited environment for the period of 2001-2020. This is achieved via a combination of the results obtained from three statistical measures: i) monthly $90^{th}$ percentile of the daily precipitation (Fig. 3a), ii) an experimentally-based precipitation frequency occurrence with a threshold of ≥10 mm/day (Fig. 3b) – after examining other thresholds such as ≥5 mm/day and ≥20 mm/day (see Figs. S1 and S2), and iii) monthly rainfall standard deviation SD estimates (Fig. 3c). These methods were calculated using a set of statistical functions, described in the follow-up subsection (see Table 1). Therefore, using the above-mentioned approach developed in this study, the Sinai's wet months are determined from October to March, defined as wet-period, and its dry months from April to September, defined as dry period.
=============================================

[Figure]

**Figure 001.** Frequency occurrence of the monthly precipitation events for the climatology period of 2001-2020 (7305 days) over the Sinai Peninsula: **left)** with a threshold of ≥9 mm/day, **right)** with a threshold of ≥11 mm/day. Units are frequency in days.

[Figure]

**Figure 3.** A multi-statistical analysis of the precipitation in a monthly basis: a) the 90[th] percentile of rainfall climatology, b) frequency occurrence of the rainfall events with a threshold of ≥10 mm/day, and c) grid-based standard deviation SD estimates of the rainfall for the period of 2001-2020 (240 months) over the Sinai Peninsula.

5) Lines 198-204: Are these numbers in trends and slopes statistically significant at a certain level of significance (say 95% or 99%)? Did authors perform any test to identify some kind of statistical significance like bootstrapping? If not, I suggest performing such tests to better aid the readers about the significance of these numbers.

**Response:** The statistical tests for significance of the regression model/trends are typically performed for the time-series dataset (between two variables), which can be determined by e.g. $r^2$-values along with the p-values (or t-test). As such, if $r^2$ is typically >0.6 with p <0.05 in regressions, then the trend is regarded as statistically significance and meaningful. However, our trend of slope (Fig.2: now Fig. S8 in the revised supplement) is estimated using an *anomaly-based approach* (not timeseries in a regression model). Here, we detected the precipitation anomalies (annual/seasonal) with a *Mean* function, meaning that the long-term mean (average) of each rainfall data was calculated; then it was subtracted from each year/season precipitation values to estimate the anomalies (i.e. anomaly equals individual values of each year/season minus long-term mean value), and finally the trend of slopes of those anomalies was drawn using the common least-squares fitting process. Therefore, our trend of slope represents the rate at which change occurs over time. If the slope has a positive value, the rainfall rate is increasing (-or the wetter condition). If it is negative, the rainfall rate is decreasing (-or the drier condition). In that figure, we interpreted the trend of slope to mean that, on average, the rainfall rate is changed by the slope value each year/season over the past two decades (2001-2020). Therefore, we deliberately avoided to use the terms such as ''significant trend'' -or ''statistically significance'', related to that figure.

Indeed, for the anomaly-based approach and its trend of slopes it is not feasible to perform any kind of statistical significances like Bootstrapping; however, we did perform *Bootstrapping Confidence Interval* for the original datasets of seasonal and annual precipitation climatology (20-years) on the selected sites across the Sinai. The results are given in Table S1 in the revised supplement data – also attached below.

However, since we now applied the EOF-based analysis (Fig. 4 in the revised manuscript, also shown below) suggested by the other respected reviewer, we decided to move the temporal site-scale anomaly-based analysis/figure into the supplement data (Fig. S8). It is good mentioning that, the results of the site-scale anomaly-based analysis are in good agreement with those of the grid-scale EOF-based spatiotemporal analysis in the Sinai Desert.

**Table S1.** The 95% and 99% bootstrapped confidence interval (CI) for the Mean and Median values of the original dataset (mean seasonal and annual for 20-years: 2001-2020) for the selected sites across the Sinai (anomaly-based analysis in Fig. S8), see Fig. 1 for the locations. For this analysis, 300 bootstrapped samples were generated each with a sample size of n=10.

| | North-site | | | Middle-site | | | South-site | | |
| --- | --- | --- | --- | --- | --- | --- | --- | --- | --- |
| | Winter | Autumn | Annual | Winter | Autumn | Annual | Winter | Autumn | Annual |
| *Average precipitation (mm)* | *68.6* | *18.5* | *28.4* | *22* | *6.1* | *9.3* | *9.1* | *4.8* | *5.1* |
| **95%** bootstrapped CI for **Mean** value of original dataset | 79.8 | 24 | 31.9 | 30.8 | 9.1 | 11.2 | 14.3 | 9.4 | 6.9 |
| **99%** bootstrapped CI for **Mean** value of original dataset | 85.5 | 26 | 33.4 | 35.4 | 10.3 | 12 | 16.2 | 10.8 | 7.3 |
| **95%** bootstrapped CI for **Median** value of original dataset | 88.3 | 26.1 | 31.4 | 23.9 | 7.5 | 10.5 | 11.7 | 5.3 | 7.5 |
| **99%** bootstrapped CI for **Median** value of original dataset | 90.4 | 29.1 | 36.1 | 29.1 | 9.7 | 12 | 17.5 | 8.2 | 8.3 |

6) Figure 3 and its analysis: What is the standard deviation of each month? While performing analysis of extreme events, knowledge of standard deviation is very important for each bar shown in these plots. Right now, I am not sure if I see any major differences between different months shown here.

**Response:** Figure 3 (now Fig. 5 in the revised manuscript) represents the monthly *ratios* of precipitation across the Sinai region in %; and it doesn't provide standard deviation SD for each month. Following your suggestion, we now added the SD values in a monthly basis for each site. Further, we also did estimate the grid-based monthly SD values for the entire Sinai region; the results are shown in Fig. 3c, see above -- SD method together with percentile -and frequency was used to develop our ''multi-statistical-approach'' to determine/split the wet -and dry months in the Sinai Desert.

It is noted that, it might be not that surprising if not major differences among different months of the sites are observed. Unlike the complex/large mountain areas like alpine/pre-alpine regions in which the climatic variables in particular precipitation may significantly vary in a short distance horizontally -or vertically, this is however not much the case for an (hyper)arid/homogeneous region like the Sinai Desert as a pretty uniform and small area from a regional/global climate perspective. Therefore, figure 5 shows that, among others, *temporally* the Sinai receives the highest monthly-precipitation-ratios in the winter months for the entire region; though *spatially* the northern parts receive a higher magnitude compared to the southern Sinai.

[Figure]

**Figure 5.** Monthly precipitation regime: (*a*) ratio of monthly sum precipitation to the annual total precipitation (%), where only ratios >20% are plotted for each month; panels (*b-g*) indicate the monthly ratios (January to December) for the selected sites; and panel (*h*) represents the standard deviation estimates (mm/month) in a monthly basis for each site shown in panel (*a*) across the Sinai Peninsula for the climatology period of 2001-2020. It is also noted that in the panel *a*, monthly ratios from April to September (colored in black in the legend) are below 20%, thus not plotted here, but full ratios (%) are illustrated in Fig. S9 in a monthly basis. In addition to the panel (*h*), full grid-based standard deviation estimate for the entire Sinai in a monthly basis is also represented in Fig. 3c.

7) Lines 225-226: When you say that "chosen sites do vary in terms of magnitude and trends", I recommend mentioning that how much do they vary actually quantitatively? Its very important to quantify these differences rather than just performing a qualitative analysis.
**Response:** Thank you for the suggestion. We now added quantitative values to highlight the differences among the sites across the study area in the revised manuscript.

8) Figure 4: Are these points statistically significant throughout the map? I am not sure if I can totally rely on these numbers without knowing the spatial statistical significance. Therefore, I recommend performing a significance test to identify which points on the map are statistically significant.
**Response:** Figure 4 (now Fig. 2a-c in the revised manuscript) displays the spatial patterns/distribution of the satellite precipitation events Sinai-wide in different time-scales: *a*) annual mean climatology, and *b, c*) records of the wettest month -and day, respectively. Ideally, tests for the statistical significance are typically used to find out what is the probability that a (meaningful) relationship -or correlation exists between two variables especially for the model observations. In that map there is only one variable of precipitation.
As you know, statistically significance tests like Bootstrapping could be performed for a singly variable; but, basically that method is applied to construct confidence interval for a statistic when the sample size is very small (e.g. from a lab work), and its underlying distribution is pretty unknown (e.g. form a model output). However, this is not really the case for remote-sensing satellite-based precipitation data, shown in Figure 2a-c (RS-data are nowadays used as *reference* e.g. to validate the model performance).

However, the statistical methods like the grid-based standard deviations estimated in this study (see Fig. 3c) could further help to understand the variations in the Sinai's precipitation spatial patterns. In Figure 2, the GPM-data and extreme indexes had been displayed both as shading and contours; however, we now removed the contours. Please also note that Figure 6 (climate extreme indices) is now attached to Figure 2d-f, due to the new subsection added in the revised manuscript i.e. (3.1.1 *The precipitation spatial patterns and extreme indices*), see Figure 2 below.

[Figure]

**Figure 2.** The precipitation spatial patterns and extreme indices: a) climatology map of mean annual precipitation (2001-2020); b) the wettest month i.e. March 2020 (out of 240 months), c) the wettest day i.e. March 12, 2020 (out of 7305 days); as well as extreme daily precipitation indices with a threshold of ≥1mm/day: d) simple daily intensity index (SDII), e) consecutive dry days (CDD) and f) wet days index (RR1) for the period of 2001-2020 over the Sinai Peninsula.

9) Section 3.2.2: I suggest not using strong words such as "a strong association is realized" as correlation is not causation. So be careful in using such sentences in your manuscript.
**Response:** It is done in the revised manuscript.

10) Section 4, Discussion: I do agree with authors' interpretation and schematic in Figure 12 depicting the primary mechanisms responsible for extreme rainfall events. However, as the authors mentioned that they observed low correlation values with atmospheric state variables which could be due to a number of factors of course. I am reiterating that correlation is not always causation and thus if the authors really wish to establish causality between atmospheric state and rainfall, I suggest using causual discovery methods such as PC and LINGAM methods. I suggest following this book if they are interested in causal discovery methods: https://matheusfacure.github.io/python-causality-handbook/landing-page.html
**Response:** Thank you for the suggestion; but this is out of the scope of our research. It can be considered in future studies though. However, it seems that the suggested methods are mostly applied in humanities, but not much in physical sciences; this is what we found by their examples (e.g. tablets for students…) made to conclude '*'association is not causation''*.

However, we believe that the spatial-correlation approach presented in this study was reasonably capable of capturing the complex dynamical relationships/correlations (but not necessarily causations) between the rainfall and atmospheric variables over the eastern Mediterranean region. Indeed, the condensation process and precipitation event are truly complicated processes to make. Basically, it doesn't have a single driver, but driven by several variables combined; and yet it varies from one region to another. Thus, it might be not that surprising to get a bit lower correlation amongst them for the Sinai's case over the eastern Mediterranean region; nevertheless, possible reasons for that are discussed in the manuscript. Therefore, our method presented in this study (which, could be one of the first-analyses in atmospheric science) can be also examined in other regions with a different regional climate from a synoptic/dynamic perspective.

11) Line 574: I am not sure if I understand what the authors mean by "spatially dependency". I suggest explaining it in a bit more detail.
**Response:** The ''spatialy dependency'' simply means, the spatial distribution of the precipitation varies across Sinai region; and thus, it shows a spatial dependency. However, to avoid misunderstanding, we now removed that term in the revised manuscript.